

# Sea-level response to melting of Antarctic ice shelves on multi-centennial time scales with the fast Elementary Thermomechanical Ice Sheet model (f.ETISh v1.0)

Frank Pattyn[1]

[1]Laboratoire de Glaciologie, Department of Geosciences, Environment and Society, Université libre de Bruxelles, Av. F.D. Roosevelt 50, B–1050 Brussels, Belgium

*Correspondence to:* Frank Pattyn (fpattyn@ulb.ac.be)

**Abstract.** The magnitude of the Antarctic ice sheet's contribution to global sea-level rise is dominated by the potential of its marine sectors to become unstable and collapse as a response to ocean (and atmospheric) forcing. This paper presents Antarctic sea-level response to sudden atmospheric and oceanic forcings on multi-centennial time scales with the newly developed *fast Elementary Thermomechanical Ice Sheet* (f.ETISh) model. The f.ETISh model is a vertically integrated hybrid ice sheet/ice
shelf model with an approximate implementation of ice sheet thermomechanics, making the model two-dimensional. Its marine boundary is represented by two different flux conditions, coherent with power-law basal sliding and Coulomb basal friction. The model has been compared to a series of existing benchmarks.

Modelled Antarctic ice sheet response to forcing is dominated by sub-ice shelf melt and the sensitivity is highly dependent on basal conditions at the grounding line. Coulomb friction in the grounding-line transition zone leads to significantly higher
mass loss in both West and East Antarctica on centennial time scales, leading to 2 m sea level rise after 500 year for a moderate melt scenario of 20 m a$^{-1}$ under freely-floating ice shelves, up to 6 m for a 50 m a$^{-1}$ scenario. The higher sensitivity is attributed to higher driving stresses upstream from the grounding line.

Removing the ice shelves altogether results in a disintegration of the West Antarctic ice sheet and (partially) marine basins in East Antarctica. After 500 years, this leads to a 4.5 m and a 12.2 m sea level rise for the power-law basal sliding and Coulomb
friction conditions at the grounding line, respectively. The latter value agrees with simulations by DeConto and Pollard (2016) over a similar period (but with different forcing and including processes of hydrofracturing and cliff failure).

The chosen parametrizations make model results largely independent of spatial resolution, so that f.ETISh can potentially be integrated in large-scale Earth system models.

## 1 Introduction

Projecting future sea-level rise requires ice sheet models capable of exhibiting complex behaviour at the contact of the ice sheet with the atmosphere, subglacial environment and the ocean. The majority of these interactions demonstrate non-linear behaviour due to feedbacks, leading to self-amplifying ice mass change. For instance, surface mass balance interacts with ice sheets through a powerful melt–elevation feedback, invoking non-linear response as a function of equilibrium line altitude, such



as a positive feedback on ablation that can be expected as the ice-sheet surface becomes lower (Levermann and Winkelmann, 2016). This feedback is also the main reason for the threshold behaviour of the Greenland ice sheet on multi-millennial time scales (e.g., Ridley et al., 2010). Typical for these self-amplifying effects is that they work both ways: the melt–elevation feedback equally allows for ice sheets to grow rapidly once a given threshold in positive accumulation is reached, resulting in hysteresis (Weertman, 1976).

Another powerful feedback relates to the contact of ice sheets (especially marine ice sheets with substantial parts of the bedrock lying below sea level) with the ocean. Mercer (1978) and Thomas (1979) identified marine ice sheet instability for ice sheets where the bedrock dips deeper inland from the grounding line (retrograde bed slopes), so that increased (atmospheric/oceanic) melting leads to recession of the grounding line. This would result in the glacier becoming grounded in deeper water with greater ice thickness. Since ice thickness at the grounding line is a key factor in controlling ice flux across the grounding line, thicker ice grounded in deeper water would result in floatation, increased ice discharge, and further retreat within a positive feedback loop. Early numerical ice sheet models failed to reproduce this feedback due to the lack of physical complexity (e.g., neutral equilibrium; Hindmarsh, 1993) and the poor spatial resolution to resolve the process of grounding line migration (Vieli and Payne, 2005; Pattyn et al., 2006). A major breakthrough was provided by an analysis of grounding line dynamics based on boundary layer theory (Schoof, 2007a, b, 2011), mathematically confirming the earlier findings by Weertman (1974) and Thomas (1979), i.e. that grounding line positions are unstable on retrograde bedrock slopes in absence of (ice shelf) buttressing. Schoof (2007a) showed that numerical ice-sheet models need to evaluate membrane stresses across the grounding line, hence resolving them on a sufficiently fine grid of less than a kilometre, which was further confirmed by two ice sheet model intercomparisons (Pattyn et al., 2012, 2013). Since then several marine ice sheet models of the Antarctic ice sheet have seen the light, with varying ways of treating the grounding line, i.e. by increasing locally spatial resolution at the grounding line (Favier et al., 2014; Cornford et al., 2015), by making use of local interpolation strategies at the grounding line (Feldmann et al., 2014; Feldmann and Levermann, 2015; Golledge et al., 2015; Winkelmann et al., 2015) or by parametrizing grounding line flux based on boundary layer theory (Pollard and DeConto, 2009; Pollard et al., 2015; Ritz et al., 2015; DeConto and Pollard, 2016).

Other feedbacks relate ice sheet dynamics to basal sliding through thermo-viscous instabilities, which may lead to limit-cycle behaviour in ice sheets (Payne, 1995; Pattyn, 1996) as well as ice stream development in absence of strong basal topographic control (Payne and Dongelmans, 1997; Payne et al., 2000; Hindmarsh et al., 2009). More elaborate subglacial water flow models have since been developed, exhibiting similar feedback mechanisms in ice discharge (Schoof, 2010). For marine portions of ice sheets, the major subglacial constraint is governed by till deformation and observations have led to new insights in subglacial till deformation based on Coulomb friction controlled by subglacial water pressure (Tulaczyk et al., 2000a, b). In contact with the ocean, subglacial water pressure may therefore stem from the depth of the bed below sea level, which led to new characterizations of grounding line dynamics (Tsai et al., 2015).

In this paper, I present a new ice sheet model that reduces the three-dimensional nature of ice sheet flow to a two-dimensional problem, while keeping the essential (or elementary) characteristics of ice sheet thermomechanics and ice stream flow. Furthermore, a number of non-linear numerical problems have been linearised in order to increase both numerical stability and improve




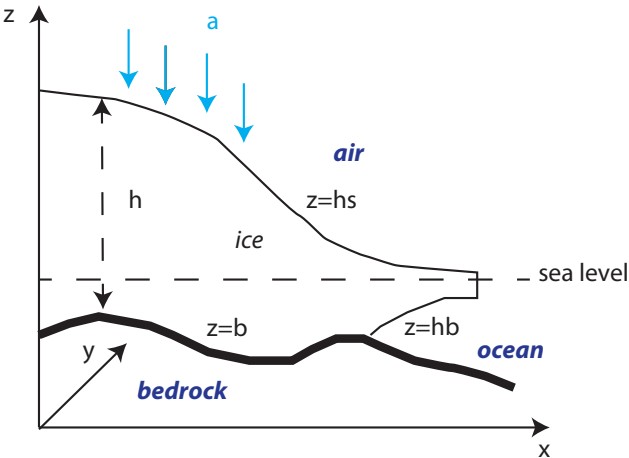

**Figure 1.** General Cartesian geometry of the f.ETISh model.

computational speed, while making sure that the processes modelled are preserved to the level of accuracy needed. Finally, processes controlling grounding line motion are adapted in such a way that they can be represented at coarser resolutions. This way, the model can more easily be integrated within computational-demanding Earth-system models. A novel grounding-line algorithm based on the zero effective pressure conditions reigning at the contact with the ocean has been implemented, which

leads to a more sensitive grounding-line response, without necessarily taking into account other mechanisms of accelerating mass loss, such as ice-cliff failure and hydro-fracturing (Pollard et al., 2015; DeConto and Pollard, 2016).

I start by giving a detailed overview of the model and its components. The initialisation procedure for the Antarctic ice sheet is then given, and finally, the sensitivity of the Antarctic ice sheet to sudden atmospheric and ocean warming is presented on centennial time scales. The appendices further describe results of known benchmarks for grounded ice flow (Huybrechts et al.,

1996; Payne et al., 2000), floating ice shelves (MacAyeal et al., 1996; Rommelaere and Ritz, 1996), and marine ice sheet dynamics (Pattyn et al., 2012).

## 2   Model description

The model consists of diagnostic equations for ice velocities, and three prognostic equations for the temporal evolution of ice thickness, ice temperature, and bedrock deformation beneath the ice. Prescribed boundary fields are equilibrium bedrock

topography, basal sliding coefficients, geothermal heat flux, and sea level. Present-day mean surface air temperatures and precipitation are derived from data assimilation within climate models. Ablation can be determined from a Positive Degree-Day model. A list of model symbols is provided in Tables 1–3. A general overview of the Cartesian geometry used is given in Fig. 1.





| Symbol | Description | Units | Value |
|---|---|---|---|
| $\dot{a}$ | Surface mass balance (SMB) | $\mathrm{m\,a^{-1}}$ | |
| $A$ | Glen's flow law factor | $\mathrm{Pa^{-n}\,a^{-1}}$ | |
| $A_b$, $A_b'$ | Basal sliding factor in power-law sliding | $\mathrm{Pa^{-m}\,m\,a^{-1}}$ | |
| $A_{\mathrm{froz}}$ | Basal sliding factor for frozen conditions | $\mathrm{Pa^{-m}\,m\,a^{-1}}$ | $10^{-10}$ |
| $b$ | Bedrock elevation | m | |
| $b_f$ | Buttressing factor | | 0–1 |
| $c_p$ | Specific heat of ice | $\mathrm{J\,kg^{-1}\,K^{-1}}$ | 2009 |
| $c_{po}$ | Specific heat of seawater | $\mathrm{J\,kg^{-1}\,K^{-1}}$ | 3974 |
| $C_r$ | Calving rate | $\mathrm{m\,a^{-1}}$ | |
| $C_s$ | Friction coefficient in Schoof (2007a) | $\mathrm{Pa\,m^{-m_s}\,s^{m_s}}$ | $(A_b'/\mathrm{spy})^{-m_s}$ |
| $c_0$ | Till cohesion | Pa | 0 |
| $d$ | Diffusion coefficient of grounded ice sheet flow | $\mathrm{m^2\,a^{-1}}$ | |
| $D$ | Flexural rigidity of lithosphere | N m | $10^{25}$ |
| $E_f$ | Adjustment factor in Arrhenius equation | | 0.035–1 |
| $F_{\mathrm{melt}}$ | Adjustment factor for sub-shelf melt rates | | 1–8 |
| $f_g$ | Fractional area of shelf grid cell in contact with bed | | 0–1 |
| $f_s$ | Scaling term for strain heating | | |
| $g$ | Gravitational acceleration | $\mathrm{m\,s^{-2}}$ | 9.81 |
| $G$ | Geothermal heat flux | $\mathrm{W\,m^{-2}}$ | |
| $h$ | Ice thickness | m | |
| $h_b$ | Bottom of ice sheet/ice shelf | m | |
| $h_e$ | Subgrid ice thickness on ice shelf edge | m | |
| $h_f$ | Ice thickness in effective viscosity | m | |
| $h_g$ | Interpolated ice thickness at grounding line | m | |
| $h_{\mathrm{max}}$ | Maximum neighbouring ice thickness | m | |
| $h_s$ | Ice sheet surface | m | |
| $h_w$ | Water column thickness under ice shelf | m | |
| $K$ | Thermal conductivity | $\mathrm{J\,m^{-1}\,s^{-1}\,K^{-1}}$ | 2.1 |
| $L$ | Latent heat of fusion | $\mathrm{J\,kg^{-1}}$ | $3.35 \times 10^5$ |
| $L_w$ | Flexural length scale of the lithosphere | | |
| $m$ | Exponent in basal sliding law | | 2 |
| $m_s$ | Basal sliding exponent in Schoof (2007a) | | $1/m$ |
| $M$ | Basal melting rate under ice shelves | $\mathrm{m\,a^{-1}}$ | |
| $n$ | Glen's flow law exponent | | 3 |
| $n_x$, $n_y$ | Outward pointing normal vectors in $x$ and $y$ | | |
| $P$ | Precipitation rate (accumulation) | $\mathrm{m\,a^{-1}}$ | |

**Table 1.** Model symbols, units and nominal values





| Symbol | Description | Units | Value |
|---|---|---|---|
| $p_w$ | Subglacial water pressure | Pa | |
| $P_w$ | Point load on bedrock | | |
| $q$ | Exponent in Coulomb friction law | | 0–1 |
| $q_b$ | Bedrock load | Pa | |
| $q_g$ | Ice flux at the grounding line | $\mathrm{m^2\,a^{-1}}$ | |
| $Q_o$ | Numerical coefficient in Tsai et al. (2015) | | 0.61 |
| $r$ | Scaling factor in sliding law | | 0–1 |
| $R$ | Gas constant | $\mathrm{J\,kg^{-1}\,mol^{-1}}$ | 8.314 |
| $S$ | Surface melt rate | $\mathrm{m\,a^{-1}}$ | |
| $S_o$ | Ocean salinity | psu | 35 |
| spy | Seconds per year | $\mathrm{s\,a^{-1}}$ | 31,556,926 |
| $T$ | Mean ice column temperature | K | |
| $T^{\mathrm{eq}}$ | Steady-state temperature | K | |
| $T_{fo}$ | Ocean freezing temperature | K | 271.03 |
| $T_m$ | Pressure melting temperature | K | |
| $T_{oc}$ | Ocean temperature | °C | |
| $T_r$ | Temperature at which basal sliding starts | °C | |
| $T_s$ | Surface temperature | K | |
| $T^{\star}$ | Homologous temperature | K | |
| $\Delta T$ | Background temperature forcing | °C | |
| $\delta T$ | Scaling factor in mass balance forcing | °C | 10 |
| $u$ | Horizontal ice velocities in $x$ direction | $\mathrm{m\,a^{-1}}$ | |
| $u_b$ | Basal velocity in $x$ direction | $\mathrm{m\,a^{-1}}$ | |
| $u_g$ | Velocity at the grounding line (Schoof, 2007a; Tsai et al., 2015) | $\mathrm{m\,a^{-1}}$ | |
| $u_0$ | Limit velocity in Coulomb friction law | $\mathrm{m\,a^{-1}}$ | 100 |
| $v$ | Horizontal ice velocities in $y$ direction | $\mathrm{m\,a^{-1}}$ | |
| $v_b$ | Basal velocity in $y$ direction | $\mathrm{m\,a^{-1}}$ | |
| $\boldsymbol{v}$ | Vertical mean horizontal velocity | $\mathrm{m\,a^{-1}}$ | |
| $\boldsymbol{v}_b$ | Horizontal basal velocity | $\mathrm{m\,a^{-1}}$ | |
| $\boldsymbol{v}_d$ | Horizontal deformational velocity | $\mathrm{m\,a^{-1}}$ | |
| $w_b$ | Lithospheric deflection | | |
| $w_c$ | Weighting factor in calving law | | 0–1 |
| $w_p$ | Response to point load on bedrock | | |
| $x, y$ | Orthogonal horizontal coordinates | m | |
| $z$ | Vertical elevation, increasing upwards from reference plane | m | |
| $z_{\mathrm{sl}}$ | Sea level elevation | m | 0 |

**Table 2.** Model symbols, units and nominal values (continued)





| Symbol | Description | Units | Value |
|---|---|---|---|
| $\beta^2$ | Basal friction coefficient | | |
| $\beta_0$ | Inverse of Péclet number | | |
| $\gamma$ | Atmospheric lapse rate | $^\circ$C m$^{-1}$ | 0.008 |
| $\gamma_T$ | Thermal exchange velocity | m s$^{-1}$ K$^{-1}$ | $5 \times 10^{-7}$ |
| $\Delta$ | Grid cell size, equal in $x$ and $y$ directions | m | |
| $\dot{\varepsilon}_{xx}, \dot{\varepsilon}_{yy}$ | Normal strain rate in $x$ and $y$ direction | a$^{-1}$ | |
| $\dot{\varepsilon}_0$ | Minimum strain rate in effective viscosity | a$^{-1}$ | $10^{-20}$ |
| $\eta$ | Effective viscosity | Pa a | |
| $\kappa$ | Thermal diffusivity | m$^2$ s$^{-1}$ | $1.1487 \times 10^{-6}$ |
| $\lambda_p$ | Scaling factor in pore water pressure | | |
| $\rho_b$ | Bedrock density | kg m$^{-3}$ | 3370 |
| $\rho_i$ | Ice density | kg m$^{-3}$ | 910 |
| $\rho_w$ | Sea water density | kg m$^{-3}$ | 1028 |
| $\omega$ | Scaled vertical velocity | | |
| $\phi$ | Till friction angle | deg | |
| $\phi_{\min}$ | minimum till friction angle | deg | 8–12 |
| $\phi_{\max}$ | maximum till friction angle | deg | 30 |
| $\sigma_b$ | Standard deviation of bedrock variability | | |
| $\Theta$ | Buttressing at grounding line | | $[0, 1]$ |
| $\theta$ | Ice temperature | K | |
| $\theta_b$ | Basal temperature | K | |
| $\theta_b^s$ | Basal temperature of the ice shelf | K | |
| $\tau_b$ | Basal drag | Pa | |
| $\tau_c$ | Coulomb stress | Pa | |
| $\tau_d$ | Driving stress | Pa | |
| $\tau_f$ | Free-water tensile stress | Pa | |
| $\tau_{xx}, \tau_{yy}$ | Longitudinal stress in $x$ and $y$ | Pa | |
| $\tau_t$ | Relaxation time for temperature | a | |
| $\tau_w$ | Relaxation time for lithospheric response | a | 3000 |
| $\zeta$ | Scaled vertical coordinate | | $[0, 1]$ |

**Table 3.** Model symbols, units and nominal values (continued)





For the coupled ice sheet/ice shelf system the surface elevation $h_s$ is defined as

$$h_s = \max\left[ b + h, \left(1 - \frac{\rho_i}{\rho_w}\right) h + z_{\mathrm{sl}} \right], \tag{1}$$

where $h$ is the ice thickness, $b$ is the bedrock elevation, $z_{\mathrm{sl}}$ is the sea-level height with respect to the chosen datum, $\rho_i$ and $\rho_w$ are the ice and seawater density, respectively. It follows that the bottom of the ice sheet equals $h_b = h_s - h$, and that $h_b = b$
holds for the grounded ice sheet.

### 2.1 Ice velocities

#### 2.1.1 Approximations

The ice sheet/ice shelf model has several modes of operation, depending on the boundary conditions that are applied. The most elementary flow regime of the grounded ice sheet is according to the Shallow-Ice approximation (SIA; Hutter, 1983), extended
with either a Weertman-type (or power-law) function or a linear/plastic Coulomb friction law for basal sliding. Ice shelf flow is governed by the Shallow-Shelf approximation (SSA; Morland, 1987; MacAyeal, 1989), defined by zero basal drag and extended by a water-pressure condition at the seaward edge. The transition between both systems is given by a flux-condition at the grounding line (Pollard and DeConto, 2009, 2012a), either derived from boundary layer theory based on SSA (SGL; Schoof, 2007a) or given by a flux-condition based on Coulomb friction at the grounding line (TGL; Tsai et al., 2015).
A second mode of operation is the hybrid mode, in which the flow regime of the grounded ice sheet is governed by a combination of SIA, responsible for ice-deformational flow, and SSA for basal sliding (Bueler and Brown, 2009; Martin et al., 2011; Winkelmann et al., 2011). The hybrid model can be used in combination with power-law sliding or linear/plastic Coulomb friction underneath the ice sheet. All components of the flow model are detailed in the sections below.

#### 2.1.2 Shallow-Ice Approximation (SIA)

The Shallow-Ice approximation (SIA; Hutter, 1983) is commonly used in ice sheet modelling. This approximation is valid for ice sheets of small aspect ratios $h \ll L$, where $L$ is the horizontal length scale of the ice sheet domain, and further characterized by a low curvature and low sliding velocities. The approximation is, however, not valid near grounding lines nor for ice shelf flow, for which other approximations are applied (see below). According to SIA, the vertical mean horizontal velocity in an ice sheet is given by

$$\boldsymbol{v}_{\mathrm{SIA}} = \boldsymbol{v}_b + \frac{2A}{n+2} h \tau_d^n, \tag{2}$$

where $\tau_d = -\rho_i g h \nabla h_s$ is the driving stress, $A$ is the flow parameter in Glen's flow law (with $n = 3$), $\boldsymbol{v}_b = (u_b, v_b)$ is the basal sliding velocity and $\boldsymbol{v}_{\mathrm{SIA}} = (u, v)$ is the vertical mean horizontal velocity according to SIA. The flow parameter $A$ is a function of ice temperature (see Sect. 2.4). The main advantage of SIA is that the velocity is completely determined from the local ice-sheet geometry.




### 2.1.3 Hybrid Shallow-Shelf/Shallow-Ice approximation (HySSA)

The flow velocity in an ice shelf or an ice stream characterized by low drag is derived from the Stokes equations (Stokes, 1845) by neglecting vertical shear terms and by integrating the force balance over the vertical. The resulting equations are (Morland, 1987; MacAyeal, 1989):

$$
\quad 2\frac{\partial}{\partial x}\left(2\eta h\frac{\partial u}{\partial x}+\eta h\frac{\partial v}{\partial y}\right)+\frac{\partial}{\partial y}\left(\eta h\frac{\partial u}{\partial y}+\eta h\frac{\partial v}{\partial x}\right)-\tau_{b_x}
$$
$$
=-\tau_{d_x}, \tag{3}
$$
$$
2\frac{\partial}{\partial y}\left(2\eta h\frac{\partial v}{\partial y}+\eta h\frac{\partial u}{\partial x}\right)+\frac{\partial}{\partial x}\left(\eta h\frac{\partial v}{\partial x}+\eta h\frac{\partial u}{\partial y}\right)-\tau_{b_y}
$$
$$
=-\tau_{d_y}, \tag{4}
$$

where

$$
\quad \eta = \frac{A^{-1/n}}{2}\left[\left(\frac{\partial u}{\partial x}\right)^2+\left(\frac{\partial v}{\partial y}\right)^2+\frac{\partial u}{\partial x}\frac{\partial v}{\partial y}+\right.
$$
$$
\left. \frac{1}{4}\left(\frac{\partial u}{\partial y}+\frac{\partial v}{\partial x}\right)^2+\dot{\varepsilon_0}^2\right]^{(1-n)/2n}, \tag{5}
$$

and where $\tau_{d_x}=\rho_i gh(\partial h_s/\partial x)$ (similar for $\tau_{d_y}$). $\dot{\varepsilon}_0=10^{-20}$ is a small factor to keep $\eta$ finite, hence to prevent singularities when velocity gradients are zero. For the ice shelf, $\tau_b=0$, while for the grounded ice sheet the basal drag is a function of the friction at the base. The SSA stress-equilibrium equations (3) and (4) require boundary conditions to be specified along the contour which defines the boundary to the ice-shelf domain, which is taken as the edge of the computational domain, irrespective of whether or not calving is considered. Dynamic conditions (specification of stress) are applied at this seaward edge, so that the vertically-integrated pressure balance then reads

$$
2\eta h\left[\left(2\frac{\partial u}{\partial x}+\frac{\partial v}{\partial y}\right)n_x+\frac{1}{2}\left(\frac{\partial u}{\partial y}+\frac{\partial v}{\partial x}\right)n_y\right]
$$
$$
=n_x\frac{1}{2}\rho_i gh^2\left(1-\frac{\rho_i}{\rho_w}\right), \tag{6}
$$
$$
\quad 2\eta h\left[\left(2\frac{\partial v}{\partial y}+\frac{\partial u}{\partial x}\right)n_y+\frac{1}{2}\left(\frac{\partial u}{\partial y}+\frac{\partial v}{\partial x}\right)n_x\right]
$$
$$
=n_y\frac{1}{2}\rho_i gh^2\left(1-\frac{\rho_i}{\rho_w}\right), \tag{7}
$$

where $n_x, n_y$ are the outward-pointing normal vectors in the $x$ and $y$ direction, respectively.

The ice shelf velocity field is needed for determining the effect of buttressing in the grounding line flux conditions (see below), as well as for the thickness evolution of the ice shelf. For the purpose of buttressing, velocity gradients downstream





from the grounding line are used to determine the longitudinal stretching rate, which is compared to the stretching rate of a freely-floating ice shelf to determine a so-called buttressing factor. This does not require a full solution of the non-linear system of ice shelf equations, and velocity gradients can be approximated from a linearised solution of the ice shelf equations. This is done by simplifying the effective viscosity Eq. (5) of the ice shelf, while keeping the essential strain-enhanced effect in the effective viscosity. For the flow-line case, the only non-zero strain rate is the stretching rate in the direction of the flow so that

$$\dot{\varepsilon}_{xx} = \frac{\partial u}{\partial x} = A\tau_f^n, \tag{8}$$

where

$$\tau_f = \frac{1}{2}\rho_i g h_f \left(1 - \frac{\rho_i}{\rho_w}\right). \tag{9}$$

where $h_f$ is defined by

$$h_f = \max\left[\min(h, 1000), 100\right] \tag{10}$$

in order to limit the variability of the effective viscosity, especially in areas with highly varying basal topography. Inserting Eq. (8) in Eq. (5) then results in

$$\eta = \frac{\tau_f^{1-n}}{2A}. \tag{11}$$

This way, the effective viscosity becomes independent of the velocity components, which significantly increases the calculation efficiency. Despite this approximation, the general behaviour of the flow field is only slightly affected, as is shown in Appendix D.

Both SIA and SSA velocities are combined to obtain the velocity field of the grounded ice sheet according to the hybrid model (HySSA; Bueler and Brown, 2009). While Bueler and Brown (2009) use a weighing function to ensure a continuous solution of the velocity from the interior of the ice sheet across the grounding line to the ice shelf, Winkelmann et al. (2011) have demonstrated that a simple addition still guarantees a smooth transition. Thus basal velocities for the grounded ice sheet are SSA velocities $\boldsymbol{v}_b = \boldsymbol{v}_{\text{SSA}}$ and

$$\boldsymbol{v} = \boldsymbol{v}_{\text{SIA}} + \boldsymbol{v}_{\text{SSA}} \tag{12}$$

for the velocity field in the grounded ice sheet.



### 2.1.4 Power-law basal sliding

Basal sliding is introduced as a Weertman sliding law, i.e.,

$$\boldsymbol{v}_b = A'_b |\tau_b|^{m-1} \tau_b, \tag{13}$$

where $\tau_b$ is the basal shear stress ($\tau_b \sim \tau_d$ for SIA), $A'_b$ is a basal sliding factor, and $m$ is the basal sliding law exponent. The
basal sliding factor $A'_b$ is temperature dependent and allows for sliding within a basal temperature range between -3 and $0°$C.
It further takes into account sub-grid sliding across mountainous terrain (Pollard et al., 2015):

$$A'_b = (1-r)A_{\mathrm{froz}} + rA_b, \tag{14}$$

where $r = \max[0, \min[1, (T^\star - T_r)/(-T_r)]]$, $A_{\mathrm{froz}}$ is the sliding coefficient in case of frozen bedrock (chosen to be very
small but different from zero to avoid singularities in the basal friction calculation), $T^\star$ is the temperature corrected for the
dependence on pressure (see Sect. 2.4.4) and $T_r = \min[-3 - 0.2\sigma_b]$, where $\sigma_b$ is the standard deviation of bedrock elevation
within the grid cell (Pollard et al., 2015). Basal sliding factors $A_b$ are either considered constant in space/time or are spatially
varying and obtained through optimization methods (see Sect. 4.1). Basal velocities in the hybrid model are defined through a
friction power law, where

$$\tau_b = \beta^2 \boldsymbol{v}_b = A'^{-1/m}_b |\boldsymbol{v}_b|^{1/m-1} \boldsymbol{v}_b. \tag{15}$$

Since Eq. (15) introduces another dependency on $\boldsymbol{v}$ in Eq. (3) and Eq. (4), the friction coefficients $\beta^2$ are approximated
by combining $|\boldsymbol{v}_b|$ with Eq. (13). Furthermore, as for 80% of the Antarctic ice sheet, driving stresses are almost completely
balanced by basal shear stress (Morlighem et al., 2013), $\tau_b \approx \tau_d$, so that

$$\beta^2 = A'^{-1/m}_b |\boldsymbol{v}_b|^{1/m-1} \approx \frac{|\tau_d|^{1-m}}{A'_b}. \tag{16}$$

### 2.1.5 Coulomb friction law

Basal friction within the HySSA equations can also be calculated based on a model for plastic till (Tulaczyk et al., 2000a).
Several variations of a basal till model can be found in the literature (Schoof, 2006; Gagliardini et al., 2007; Bueler and Brown,
2009; Winkelmann et al., 2011). Deformation of saturated till is well modelled by a plastic (Coulomb friction) or nearly plastic
rheology (Truffer et al., 2000; Tulaczyk et al., 2000a; Schoof, 2006). Its yield stress $\tau_c$ satisfies the Mohr–Coulomb relation:

$$\tau_c = c_0 + \tan\phi\,(\rho_i g h - p_w), \tag{17}$$





where the term between brackets is the effective pressure of the overlying ice on the saturated till (Cuffey and Paterson, 2010), or the ice overburden pressure minus the water pressure $p_w$, $c_0$ is the till cohesion ($c_0 = 0$ is further considered), and $\phi$ is the till friction angle. The latter can be either taken as a constant value or vary as a function of bedrock elevation (Maris et al., 2014):

$$\phi = -\phi_{\min}\frac{b}{10^3} + \left(1 + \frac{b}{10^3}\right)\phi_{\max}, \tag{18}$$

and limited by $\phi = \phi_{\min}$ for $b \leq -10^3\text{m}$ and $\phi = \phi_{\max}$ for $b \geq 0$.

The most comprehensive approach to solve for the subglacial water pressure in Eq. (17) is due to Bueler and van Pelt (2015) by considering a hydrological model of subglacial water drainage within the till. However, Martin et al. (2011) propose to relate major till characteristics to bedrock geometry and allow till friction angle and basal water pressure to be a function of the bed elevation compared to sea level. This leads to zones of weak till and saturation in subglacial basins that are well below sea level (Martin et al., 2011; Maris et al., 2014). Following their analysis, the subglacial water pressure is defined by

$$p_w = 0.96\lambda_p\rho_i gh. \tag{19}$$

Here, $\lambda_p$ is a scaling factor such that the pore water pressure is maximal when the ice is resting on bedrock at or below sea level. Below sea level, the pores in the till are assumed to be saturated with water so $\lambda_p$ is then equal to 1. The factor $\lambda_p$ is scaled with the height above sea level up until 1000 m. At and above 1000 m, $\lambda_p$ is equal to 0 (Maris et al., 2014). While there is no direct physical evidence for such water-pressure distribution in the interior of ice sheets, near grounding lines in direct contact with the ocean, subglacial water pressure of saturated till may also be approximated by (Tsai et al., 2015):

$$p_w = -\rho_w gb, \tag{20}$$

which is valid for $b < 0$, otherwise $p_w = 0$. By definition, $p_w = \rho_i gh$ at the grounding line and underneath floating ice shelves, so that the effective pressure becomes zero. Bueler and Brown (2009) consider the pore water pressure locally as at most a fixed fraction (95%) of the ice overburden pressure $\rho_i gh$. Winkelmann et al. (2011) use a fraction of 0.96, which is applied in Eq. (19).

To link Coulomb friction to basal drag, the formulation proposed by Bueler and van Pelt (2015) is opted for, where $\tau_c$ and $\boldsymbol{v}_b$ combine to determine $\tau_b$ through a sliding law, i.e.,

$$\tau_b = \tau_c\frac{\boldsymbol{v}_b}{|\boldsymbol{v}_b|^{1-q}u_0^q}, \tag{21}$$

where $0 \leq q \leq 1$, and $u_0$ is a threshold sliding speed (Aschwanden et al., 2013). The Coulomb friction law, Eq. (21), includes the case $q = 0$, leading to the purely plastic (Coulomb) relation $\tau_b = \tau_c\boldsymbol{v}_b/|\boldsymbol{v}_b|$. At least in the $q \ll 1$ cases, the magnitude of



the basal shear stress becomes nearly independent of $|\boldsymbol{v}_b|$, when $|\boldsymbol{v}_b| \gg u_0$. Equation (21) could also be written in a generic power-law form $\tau_b = \beta^2 |\boldsymbol{v}_b|^{q-1} \boldsymbol{v}_b$ with coefficient $\beta^2 = \tau_c / u_0^q$; in the linear case $q = 1$, $\beta^2 = \tau_c / u_0$ (Bueler and van Pelt, 2015).

Alternatively, both the power-law sliding law Eq. (13) and the Coulomb friction law Eq. (21) can be combined (Tsai et al., 2015; Asay-Davis et al., 2015), by taking the lowest friction value of both. Since at the grounding line basal sliding velocities are considered highest, this equally implies high basal drag in a traditional power-law sliding law. However, expressed as a basal friction law, Eq. (15) enables to derive high sliding velocities at low and near-zero basal drag. Nevertheless, power law sliding/friction still leads to a relatively sharp transition in $\tau_b$ at the grounding line (Tsai et al., 2015). Coulomb basal conditions imply that basal drag vanishes towards the grounding line, thus ensuring a smooth transition between the ice stream and ice shelf. Expressing the basal traction as

$$\tau_b = \min \left[ \beta^2 \boldsymbol{v}_b, \frac{\tau_c \boldsymbol{v}_b}{|\boldsymbol{v}_b|^{1-q} u_0^q} \right] \tag{22}$$

ensures that it is continuous (though not differentiable) across the grounding line (Asay-Davis et al., 2015). The Coulomb friction law has been implemented in f.ETISh, but substantial tests have not been carried out in the scope of this paper.

### 2.1.6 Grounding-line flux condition for power-law sliding (SGL)

Previous studies have indicated that it is necessary to resolve the transition zone/boundary layer at sufficiently fine resolution in order to capture grounding-line migration accurately (Durand et al., 2009; Pattyn et al., 2012, 2013; Pattyn and Durand, 2013; Durand and Pattyn, 2015). In large-scale models, this can lead to unacceptably small time-steps and costly integrations. Pollard and DeConto (2009, 2012a) incorporated the boundary layer solution of Schoof (2007a) directly in a numerical ice-sheet model at coarse grid resolution, so the flux, $q_g$, across model grounding lines is given by

$$q_g = \left[ \frac{A(\rho_i g)^{n+1}(1 - \rho_i/\rho_w)^n}{4^n C_s} \right]^{\frac{1}{m_s+1}} \Theta^{\frac{n}{m_s+1}} h_g^{\frac{m_s+n+3}{m_s+1}}. \tag{23}$$

This yields the vertically averaged velocity $u_g = q_g / h_g$ where $h_g$ is the ice thickness at the grounding line. $\Theta$ in Eq. (23) accounts for back stress at the grounding line due to buttressing by pinning points or lateral shear, and is defined as

$$\Theta = \frac{b_f \tau_{xx} + (1 - b_f)\tau_f}{\tau_f}, \tag{24}$$

where $\tau_{xx}$ is the longitudinal stress just downstream of the grounding line, calculated from the viscosity and strains in a preliminary SSA solution without constraints given by Eq. (23), and $\tau_f$ the free-water tensile stress defined in Eq. (9). $b_f$ is an additional buttressing factor to control the buttressing strength of ice shelves and varies between 0 (no buttressing) and 1 (full





buttressing). As in Pollard and DeConto (2012a), $C_s$ is Schoof's basal sliding coefficient and $m_s$ the basal sliding exponent, so that $C_s$ is related to the sliding coefficients $A_b'$ by $C_s = (A_b'/\mathrm{spy})^{-m_s}$, where 'spy' is the number of seconds per year and $m_s = 1/m$. Grounding-line ice thickness $h_g$ is linearly interpolated in space by estimating the sub-grid position of the grounding line between the two surrounding floating and grounded $h$-grid points. Therefore, the height above floatation is

5 linearly interpolated between those two points to where it is zero. Subsequently, the bedrock elevation is linearly interpolated to that location, and the floatation thickness of ice for that bedrock elevation and current sea level is obtained (Pattyn et al., 2006; Gladstone et al., 2010; Pollard and DeConto, 2012a). The velocity $u_g$ is then calculated at the grounding-line points and imposed as an internal boundary condition for the flow equations, hence overriding the large-scale velocity solution at the grounding line. $u_g = q_g/h_g$ is imposed exactly at the $u$-grid grounding line point when the flux $q_g$ is greater than the large-scale

sheet-shelf equation's flux at the grounding line. This is a slight variant of Pollard and DeConto (2012a).

Equation (23) applies equally to the $y$ direction, with $v_g$ and $\tau_{yy}$ instead of $u_g$ and $\tau_{xx}$. Note that spatial gradients of quantities parallel to the grounding line, which are not included in Schoof's flow-line derivation of Eq. (23), are neglected here (Katz and Worster, 2010; Gudmundsson et al., 2012; Pattyn et al., 2013). This parametrization was also found to yield results comparable to SSA models solving transient grounding line migration at high spatial resolution of the order of hundreds of

15 meters (Pattyn and Durand, 2013; Durand and Pattyn, 2015), despite the fact that Eq. (23) applies to steady-state conditions.

### 2.1.7 Grounding-line flux condition for Coulomb friction (TGL)

The grounding-line parametrization based on the boundary layer theory by Schoof (2007a) is invalid when Coulomb friction near the grounding line is considered and the effective stress tends to zero. However, Tsai et al. (2015) offers such a solution for vanishing Coulomb friction at the grounding line, and therefore independent of basal sliding coefficients:

$$20 \quad q_g = Q_o \frac{8A\left(\rho_i g\right)^n}{4^n \tan\phi} \left(1 - \frac{\rho_i}{\rho_w}\right)^{n-1} \Theta^{n-1} h_g^{n+2}, \qquad (25)$$

where $Q_o \approx 0.61$ is a numerical coefficient determined from the boundary-layer analysis. The flux in the $y$ direction is obtained in a similar fashion. As in Eq. (23), buttressing scales to the same power as $(1 - \rho_i/\rho_w)$, which is $n-1$. The performance of both flux conditions is tested in Appendix C.

The TGL flux condition can be used in conjunction with power-law basal sliding. Indeed, Tsai et al. (2015) have shown

that the crossover from Coulomb to power-law roughly occurs at stresses $\gtrsim 100$ kPa, hence the Coulomb regime occurs within $\lesssim 17$ m above the floatation height. This is a very small height difference, which implies that in most cases —with exception of ice plains— a narrow Coulomb regime exists, within a grid cell of a continental-scale model.



## 2.2 Ice thickness evolution

Ice sheet thickness evolution is based on mass conservation, leading to the continuity equation. For the general ice sheet/ice shelf system, this is written as:

$$\frac{\partial h}{\partial t} = -\frac{\partial (uh)}{\partial x} - \frac{\partial (vh)}{\partial y} + \dot{a} - M \,, \tag{26}$$

where $\dot{a}$ is the surface mass balance (accumulation minus surface ablation), and $M$ is the basal melt rate (solely underneath ice shelves, as basal melt rates underneath the ice sheet are not accounted for). The treatments of the various local ice gains or losses (surface mass balance, etc.) are described in later sections. For the grounded ice sheet, Eq. (26) is written as a diffusion equation for ice thickness (Huybrechts, 1992):

$$\frac{\partial h}{\partial t} = \frac{\partial}{\partial x}\left(d\frac{\partial (h+h_b)}{\partial x}\right) + \frac{\partial}{\partial y}\left(d\frac{\partial (h+h_b)}{\partial y}\right) + \dot{a} - M \,, \tag{27}$$

where $h_b$ is the bottom of the ice sheet (or the bedrock elevation $b$ for the grounded ice sheet).

It is also ensured that thinning due to grounding line retreat does not exceed the maximum permissible rate, using theoretical knowledge of maximum possible stresses at the grounding line that is called the 'maximum strain check'. Similar to Ritz et al. (2015), tensile stresses are ensured to not exceed those from buttressing by water alone, i.e., the free-water tensile stress, and calculate the maximum corresponding strain rate, expressed as a maximum thinning rate. The free-water tensile strain-rate then

becomes

$$h\frac{\partial u}{\partial x} = h\frac{\partial v}{\partial y} = Ah\tau_f \,. \tag{28}$$

Using the mass conservation equation (26), the condition on maximum strain rate is

$$\begin{aligned}
\frac{\partial h}{\partial t} &= \dot{a} - M - \frac{\partial (uh)}{\partial x} - \frac{\partial (vh)}{\partial y} \\
&\leq \dot{a} - M - h\frac{\partial u}{\partial x} - h\frac{\partial v}{\partial y} = \dot{a} - M - 2Ah\tau_f \,.
\end{aligned} \tag{29}$$

This is valid for $\partial h/\partial x < 0$ when $u > 0$ and $\partial h/\partial y < 0$ when $v > 0$. Ritz et al. (2015) use a slightly different prescription, but sensitivity tests showed that the extra terms in the mass conservation equation can be safely dropped, rendering the maximum strain check therefore independent of velocity gradients.





## 2.3 Calving and sub-shelf pinning

Ice-front calving is obtained from the large scale stress field (Pollard and DeConto, 2012a), based on the horizontal divergence of the ice-shelf velocities and which is similar to parametrizations used elsewhere (Martin et al., 2011; Winkelmann et al., 2011; Levermann et al., 2012). The calving rate $C_r$ is defined as

$$C_r = 30(1 - w_c) + 3 \times 10^5 \max\left(\frac{\partial u}{\partial x} + \frac{\partial v}{\partial y}, 0\right)\frac{w_c h_e}{\Delta} \tag{30}$$

where $w_c = \min(1, h_e/200)$ is a weight factor and $h_e$ is the subgrid ice thickness within a fraction of the ice edge grid cell that is occupied by ice (Pollard and DeConto, 2012a), defined by

$$h_e = \max\left[h_{\max} \times \max\left(0.25, e^{-h_{\max}/100}\right), 30, h\right] \tag{31}$$

where a minimum ice thickness of 30 m avoids too thin ice shelves. The value of $h_{\max}$ is defined as the maximum ice thickness of the surrounding grid cells (grounded or floating) that are not adjacent to the ocean (Pollard and DeConto, 2012a).The calving rate $C_r$ is then subtracted from the basal melt rate $M$ in Eq. (26).

Given the relatively low spatial resolution of a large-scale ice-sheet model, small pinning points underneath ice shelves due to small bathymetric rises scraping the bottom of the ice and exerting an extra back pressure on the ice shelf (Berger et al., 2016; Favier et al., 2016) are not taken into account. To overcome this a simple parametrization based on the standard deviation of observed bathymetry within each model cell was accounted for to introduce a given amount of basal friction of the ice shelf (Pollard and DeConto, 2012a). The fractional area $f_g$ of ice in contact with sub-grid bathymetric high is defined as (modified from Pollard and DeConto, 2012a):

$$f_g = \max\left[0, 1 - \frac{h_w}{\sigma_b}\right] \tag{32}$$

where $h_w$ is the thickness of the water column underneath the ice shelf and $\sigma_b$ is the standard deviation of the bedrock variability (see above). This factor $f_g$ is multiplied with $\beta^2$ in the basal friction. For the grounded ice sheet, $f_g = 1$; for the floating ice shelf in deeper waters, $f_g = 0$, so that the ice shelf does not experience any friction.

## 2.4 Ice temperature and rheology

Ice temperature is calculated in a semi-analytical fashion to provide an estimate of both basal temperature and the mean ice-column temperature over a given depth. The former determines regions of potential basal sliding, while the latter is employed to determine the vertically-integrated value of the flow parameter $A$ in Glen's flow law. These simplifications allow for the model to remain two-dimensional, but taking into account the basic mechanisms of major thermodynamic processes, contrary to models employing a linear temperature profile (e.g., Kavanaugh and Cuffey, 2009; Golledge and Levy, 2011). The steady-state temperature profile is a function of vertical diffusion and advection, and extended with frictional and strain heating at





the base. This is a variant of derivations due to Hindmarsh (1999) and Pattyn (2010). A solution to the horizontal advection problem was also tested (Glasser and Siegert, 2002), based on the column model due to Budd et al. (1971). However, due to the inherent simplifications, it works best when surface slopes and lapse rates are lowest (Hooke, 2005), which results in an overestimation of horizontal advection at the edges of an ice sheet, cooling down areas that are supposedly at pressure melting

point. To compensate for the lack of horizontal advection in the model, strain heating was decreased by a given fraction. Finally, a time-dependency is introduced by treating the evolution of the column-ice temperature as a relaxation equation.

### 2.4.1   Ice-sheet temperature

The steady-state diffusion–advection equation for an ice sheet near its centre (in absence of horizontal advection), is given by

$$\beta_0 \frac{\partial^2 \theta}{\partial \zeta^2} - \omega(\zeta) \frac{\partial \theta}{\partial \zeta} = 0, \tag{33}$$

where $\beta_0 = \kappa / h \dot{a}$, $\kappa = K / \rho_i c_p$ is the thermal diffusivity of ice, $K$ is the thermal conductivity, $c_p$ is the heat capacity of ice, $\theta$ is the ice temperature, $\zeta = (h_s - z)/h$ is the scaled vertical elevation, with $\zeta = 0$ at the surface and $\zeta = 1$ at the bottom of the ice sheet, and $\omega$ is the vertical velocity normalized by the surface mass balance rate, so that $\omega(\zeta = 0) = -1$. This relation has a first integral (Hindmarsh, 1999)

$$\frac{\partial \theta}{\partial \zeta} = \frac{\partial \theta_b}{\partial \zeta} \exp\left[ \frac{W(\zeta)}{\beta_0} \right] \tag{34}$$

$$W(\zeta) = \int_1^\zeta \omega(\zeta') \mathrm{d}\zeta', \tag{35}$$

where $\partial \theta_b / \partial \zeta$ is the basal temperature gradient. The scaled vertical velocity $\omega$ according to the Shallow-ice approximation is a function of the exponent of Glen's flow law (Hindmarsh, 1999):

$$\omega = -\frac{\zeta^{n+2} - \zeta(n+2) + n + 1}{n+1} \tag{36}$$

so that its integral transforms to

$$W = \frac{\zeta^{n+3} - 1}{(n+1)(n+3)} - \frac{(\zeta^2 - 1)(n+2)}{2(n+1)} + \zeta - 1. \tag{37}$$

The scaled temperature is then obtained through vertical integration of Eq. (34):

$$\theta - T_s = \frac{\partial \theta_b}{\partial \zeta} \int_1^\zeta \exp\left[ \frac{W(\zeta')}{\beta_0} \right] \mathrm{d}\zeta', \tag{38}$$





where $T_s$ is the temperature at the surface of the ice sheet. The basal boundary condition is given by

$$\frac{\partial \theta_b}{\partial \zeta} = -\frac{G + \tau_d \left(v_s + f_s v_d\right)}{K}, \tag{39}$$

where $G$ is the geothermal heat flux and the second term represents frictional heating at the base. The last term in Eq. (39) represents strain heating, where $v_d$ is the deformational velocity component ($v_d = v - v_b$). Recognizing that most of the strain heating occurs near the bed, it can be added to the geothermal heat flux (Hooke, 2005). However, to compensate for the absence of horizontal advection in the model, only a fraction $f_s \approx 0.25$ of the total strain heating amount was added. This value is determined from the EISMINT benchmark experiments (Appendix A).

### 2.4.2 Ice-shelf temperature

In ice shelves, a simple temperature model is adopted, considering the accumulation at the surface balanced by basal melting underneath an ice shelf and with only vertical diffusion and advection into play (Holland and Jenkins, 1999):

$$\theta(\zeta) = \frac{(T_s - \theta_b^s)\exp(\beta_1) + \theta_b^s - T_s \exp(\beta_2)}{1 - \exp(\beta_2)}, \tag{40}$$

where $\beta_1 = \dot{a}\zeta h/\kappa$, $\beta_2 = \dot{a}h/\kappa$, and $\theta_b^s$ is the ocean temperature at the base of the ice shelf, corrected for ice-shelf depth, i.e., $\theta_b^s = T_{oc} = -1.7 - 0.12 \times 10^{-3} h_b$ (Maris et al., 2014).

### 2.4.3 Temperature evolution

The mean column temperature $T$ is obtained by integrating $\theta$ from the base of the ice sheet to a given height in the ice column. Since most of the ice deformation is in the bottom layers of the ice sheet, the temperature closest to the bottom determines to a large extent the deformational properties. Compared to full thermomechanically-coupled ice sheet models, satisfactory results where obtained by considering a mean column temperature for the lower most 10-40% of the ice column. This fraction can also be regarded as an extra tuning parameter in an ensemble run, especially given the large uncertainties pertaining to geothermal heat flow underneath major ice sheets. The time evolution of the mean column temperature is introduced as a relaxation equation based on the Péclet number, i.e.,

$$\frac{\partial T}{\partial t} = -\frac{1}{\tau_t}\left(T - T^{\text{eq}}\right). \tag{41}$$

where $T^{\text{eq}}$ is the steady-state column temperature as calculated with the above-described procedure. Given that the Péclet number $\text{Pe} = h\dot{a}/\kappa$ is the ratio between the characteristic time scales of advection to diffusion, the time scale of each of the processes will then determine the relaxation time needed to reach a steady-state column temperature, i.e.,

$$\tau_t = \min \begin{cases} h/\dot{a} & \text{(advection)} \\ h^2/\kappa & \text{(diffusion)} \end{cases} \tag{42}$$





The main advantage of this scheme, besides being two-dimensional in nature, is that a steady-state temperature field and rheological parameters are readily obtained, reducing the initialization or spin-up time significantly. Comparison of this temperature evolution scheme with conventional three-dimensional models is given in Appendix A and B.

### 2.4.4 Thermomechanical coupling

The flow parameter $A$ and its temperature dependence on temperature are specified as in Huybrechts (1992) and Pollard and DeConto (2012a):

$$
\begin{aligned}
A &= E_f \times 5.47 \times 10^{10} \exp\left(\frac{-13.9 \times 10^4}{RT^\star}\right) \\
&\quad \text{if} \quad T^\star \geq 263.15\,\text{K}, \\
A &= E_f \times 1.14 \times 10^{-5} \exp\left(\frac{-6.0 \times 10^4}{RT^\star}\right) \\
&\quad \text{if} \quad T^\star < 263.15\,\text{K},
\end{aligned}
$$

(43)

(44)

where $T^\star = T - T_m$ is the homologous temperature, with $T_m = -8.66 \times 10^{-4}(1-\zeta)h$ the pressure melting correction and $R$ the gas constant. Units of $A$ are Pa$^{-3}$ yr$^{-1}$ corresponding to $n = 3$. The enhancement factor $E_f$ is set to 1 for the main ice sheet model, but lower for the flow of ice shelves. The ratio of enhancement factors represent differences in fabric anisotropy between grounded and ice shelf ice (Ma et al., 2010). Moreover, given the linearisation of the SSA equations, this further requires an adjustment (see Appendix D). Verification of the thermomechanical coupling scheme using a vertical mean value of $A$ is detailed in Appendix B.

### 2.5 Bedrock deformation

The response of the bedrock to changing ice and ocean loads is solved through a combined time-lagged asthenospheric relaxation and elastic lithospheric response due to the applied load (Huybrechts and de Wolde, 1999; Pollard and DeConto, 2012a). The deflection of the lithosphere is given by

$$
D\nabla^4 w_b + \rho_b g w_b = q_b ,
$$

(45)

where $D$ is the flexural rigidity of the lithosphere, and $\rho_b$ is the bedrock density. The load is then defined by

$$
q_b = \rho_i g h + \rho_w g h_w - \rho_i g h^{\text{eq}} - \rho_w g h_w^{\text{eq}} ,
$$

(46)





where $h_w$ is the ocean column thickness, and $h^{\mathrm{eq}}$ and $h_w^{\mathrm{eq}}$ are the values of ice thickness and ocean column thickness in equilibrium, respectively. Equation (45) is solved by a Green's function (Huybrechts and de Wolde, 1999). The response to a point load $P_w$ ($q_b \times$ area) versus distance from the point load $l$ is then given by

$$w_p(l) = \frac{P_w L_w^2}{2\pi D} \mathrm{kei}\left(\frac{l}{L_w}\right),$$ (47)

where kei is a Kelvin function of zeroth order (defined as the imaginary part of a modified Bessel function of the second kind), and $L_w = (D/\rho_b g)^{1/4} \approx 132$ km is the flexural length scale. For any load, the different values of the point loads $w_p$ are summed over all grid cells to yield $w_b(x,y)$. Finally, the actual rate of change in bedrock elevation is given by a simple relaxation scheme:

$$\frac{\partial b}{\partial t} = -\frac{1}{\tau_w}\left(b - b^{\mathrm{eq}} + w_b\right),$$ (48)

where $b$ is the actual bedrock elevation, $b^{\mathrm{eq}}$ is the elevation in equilibrium, and $\tau_w = 3000$ year (Pollard and DeConto, 2012a).

## 2.6 Numerical grid and solution

The ice sheet-shelf model uses a finite-difference staggered grid, where horizontal velocities $(u,v)$ are calculated on two separate staggered Arakawa C-grids, as is usual for vector fields (Rommelaere and Ritz, 1996), while diffusion coefficients for the ice-sheet equation $d$ are calculated on an Arakawa B-grid, staggered in both $x$ and $y$ direction, since these are scalar quantities (Fig. 2). The f.ETISh model is essentially two-dimensional, with variable coordinates $(x,y)$ in the plane. The ice sheet model uses no vertical coordinate, i.e., the model is vertically-integrated. However, for analytical calculations of the vertical temperature distribution a vertical grid is introduced for the purpose of local numerical integration.

The SSA velocity field Eqs. (3–4) is solved as a sparse linear system where both $u$ and $v$ component are solved as once in one matrix **A** with size $(2 \times N_x \times N_y)$ by $(2 \times N_x \times N_y)$:

$$\begin{pmatrix} A_{ux} & A_{vx} \\ A_{uy} & A_{vy} \end{pmatrix} \cdot \begin{pmatrix} u \\ v \end{pmatrix} = \begin{pmatrix} b_x \\ b_y \end{pmatrix}$$ (49)

where $N_x$, $N_y$ are the number of grid points in the $x$, $y$ direction, respectively. The submatrices $A_{ux}, A_{vx}$ contain the coefficients for the solution in the $x$ direction for $u$ and $v$, respectively. $A_{uy}, A_{vy}$ are defined in a similar way. Due to the independent nature of the effective viscosity $\eta$ on $u,v$, the solution requires no iteration. A similar solution approach is taken for solving the continuity equation for ice thickness (Payne and Dongelmans, 1997), which was favoured over an Alternating Direct Implicit scheme used in several ice-sheet models (Huybrechts, 1992; Pollard and DeConto, 2012a).

The f.ETISh model is implemented in MATLAB®. Computational improvements involved the omission of all *for*-loops by using circular shifts (with exception of the time loop), thereby optimizing the use of matrix operations. The bulk of com-



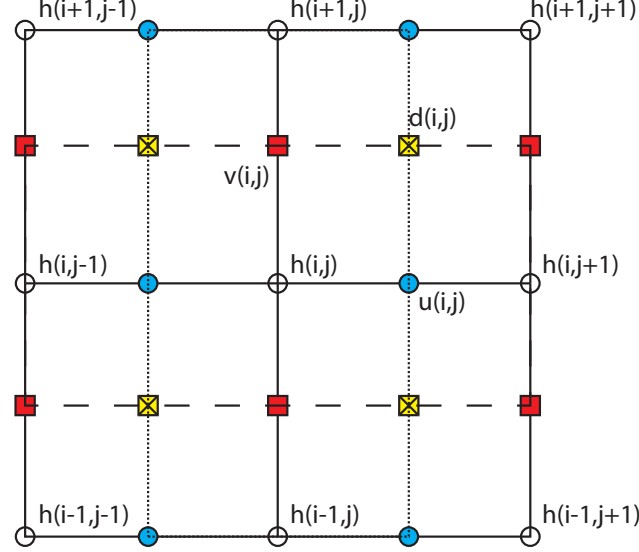

**Figure 2.** Staggered grids used in the model: the basic grid is the ice-thickness grid (shown in open circles). $u$ and $v$ velocities for the ice shelves (and ice streams) are calculated on two different staggered Arakawa C grids (filled circles and squares, respectively). Diffusion coefficients $d$ in the ice-sheet equation are solved on an Arakawa B grid (crossed squares).

putational time is devoted to the solution of the sparse matrix systems, which are natively optimized in MATLAB® using multi-threading. A preconditioned conjugate gradient method is used for solving the ice sheet/ice shelf continuity equation. The velocity field in the hybrid model is solved using a stabilized bi-conjugate gradients method, which is also preconditioned and further initialized by the velocity field solution from the previous time step. Both numerical solvers are iterative and the preconditioning limits the number of iterations to reach convergence.

The f.ETISh model is compared to other ice sheet models via a series of benchmarks, such as the EISMINT-I benchmark for isothermal ice-sheet models (Huybrechts et al., 1996, Appendix A), the EISMINT-II benchmark for thermomechanically-coupled ice sheet models (Payne et al., 2000, Appendix B), and the MISMIP experiments for marine ice-sheet models (Pattyn et al., 2012, Appendix C). Results show that the f.ETISh model is in close agreement with all of the benchmark experiments.

## 3   Input and climate forcing

### 3.1   Input data sets

For modelling the Antarctic ice sheet, the bedrock topography is based on the Bedmap2 data (Fretwell et al., 2013), from which ice thickness, present-day surface topography and grounding-line position are derived. Surface mass balance and temperatures are obtained from Van Wessem et al. (2014), based on the output of the regional atmospheric climate model RACMO2 for the period 1979–2011 and evaluated using 3234 *in situ* mass balance observations and ice-balance velocities.





For geothermal heat flux we employ a recent update of Fox-Maule et al. (2005) due to Purucker (2013). It is based on low-resolution magnetic observations acquired by the CHAMP satellite between 2000 and 2010, and produced from the MF-6 model following the same technique as described in Fox-Maule et al. (2005).

All datasets are resampled on the spatial resolution used for the experiments. The experiments shown in this paper employ a grid spacing of 25 (and in a few cases 40 or 16) km.

## 3.2 Atmospheric and ocean forcing

Atmospheric forcing is applied in a parametrized way, based on the observed fields of precipitation (accumulation rate) and surface temperature. For a change in background (forcing) temperature $\Delta T$, corresponding fields of precipitation $P$ and atmospheric temperature $T_s$ are defined by (Huybrechts et al., 1998; Pollard and DeConto, 2012a)

$$T_s = T_s^{\mathrm{obs}} - \gamma(h_s - h_s^{\mathrm{obs}}) + \Delta T, \tag{50}$$

$$P = \dot{a}^{\mathrm{obs}} \times 2^{(T_s - T_s^{\mathrm{obs}})/\delta T}, \tag{51}$$

where $\gamma = 0.008°\mathrm{C\ m}^{-1}$ is the lapse rate and $\delta T$ is $10°\mathrm{C}$ (Pollard and DeConto, 2012a). The subscript 'obs' refers to the present-day observed value. Any forcing (increase) in background then leads to an overall increase in surface temperature corrected for elevation changes according to the environmental lapse rate $\gamma$. The parametrizations of $T_s$ and $P$ can easily be replaced by values that stem from GCMs, with appropriate corrections for surface elevation (e.g., de Boer et al., 2015).

Surface melt is parametrized using a positive degree-day model (Huybrechts and de Wolde, 1999). The total amount of positive degree days (PDD) is obtained as

$$\mathrm{PDD} = \frac{1}{\sigma\sqrt{2\pi}} \int_0^A \left[ \int_0^{\overline{T}+2.5\sigma} T \exp\left( -\frac{(T-\overline{T})^2}{2\sigma^2} \right) \mathrm{d}T \right] \mathrm{d}t, \tag{52}$$

where $\sigma$ is taken as $5°\mathrm{C}$ (Reeh, 1989) and $\overline{T}$ is the mean annual temperature. The annual number of positive degree days represents a melt potential, used to melt snow and (superimposed) ice. This is determined by applying a seasonal cycle to the atmospheric temperatures with a double amplitude of $20°\mathrm{C}$, linearly increasing to $30°\mathrm{C}$ at an elevation of 3000 m, and kept at $30°\mathrm{C}$ at higher elevations (Pollard and DeConto, 2012a). The PDD melt potential is related to surface melt through a coefficient of 0.005 m of melt per degree day (Pollard and DeConto, 2012a). Although more complex schemes are often used, taking into account refreezing of percolating meltwater in the snow pack and melting of superimposed ice with different melt coefficients (Huybrechts and de Wolde, 1999), which is also confirmed by recent observations (Machguth et al., 2016), surface melt is rather limited for the present-day Antarctic ice sheet. Surface mass balance is then the sum of the different components, i.e., $\dot{a} = P - S$, where $S = 0.005 \times$ PDD is the surface melt rate.



Melting underneath the floating ice shelves is often based on parametrizations that relate sub-shelf melting to ocean temperature and ice-shelf depth (Beckmann and Goosse, 2003; Holland et al., 2008), either in a linear or a quadratic way (Martin et al., 2011; Pollard and DeConto, 2012a; de Boer et al., 2015; DeConto and Pollard, 2016). This leads to higher melt rates close to the grounding line, as the ice-shelf bottom is the lowest. The adaptation by Holland et al. (2008) and Pollard and DeConto (2012a) is implemented in f.ETISh, where the dependence on temperature difference is quadratic:

$$M = F_{\text{melt}} \frac{\rho_w c_{po} \gamma_T}{L \rho_i} |T_{oc} - T_{fo}| (T_{oc} - T_{fo}) \,, \tag{53}$$

and where $M$ is the sub-ice-shelf basal melt rate, $c_{po}$ is the specific heat capacity of the ocean, $\gamma_T$ is the thermal exchange velocity, $L$ is the latent heat of fusion, $F_{\text{melt}}$ is a predefined melt factor, depending on the potential for warm ocean currents to access the cavity beneath the ice shelf, $T_{oc}$ is the temperature of the ocean underneath the ice shelf, and $T_{fo}$ is the freezing temperature defined by Beckmann and Goosse (2003) as:

$$T_{fo} = 0.0939 - 0.057 S_o + 7.64 \times 10^{-4} h_b \,, \tag{54}$$

where $S_o$ is a mean value for the salinity of the ocean of 35 psu. For determining the melt factor $F_{\text{melt}}$ a distinction is made between protected ice shelves (Ross and Ronne-Filchner; Fig. 3) with a melt factor of $F_{\text{melt}} = 1$ and all other ice shelves with a melt factor of $F_{\text{melt}} = 8$ (Pollard and DeConto, 2012a). A similar approach has been taken by many other ice-sheet models cited in de Boer et al. (2015). The parametrized melt rate in Eq. (53) follows a quadratic function of ice shelf bottom and thus results in the highest melt rates closest to the grounding line where the ice shelf is thickest, which may not always be the case according to coupled ocean-ice shelf modelling (De Rydt and Gudmundsson, 2016). Favier et al. (2016) used different commonly-used distributions for sub-shelf melting and found significantly different grounding line transient responses. On top of this, recent observations show that the spatial variability in sub-shelf melt rates for the Antarctic ice shelves is quite large and hard to quantify by a simple parametrization (Schodlok et al., 2016). Therefore, a constant value of basal ice-shelf melt was used as a sensitivity parameter in our experiments (independent of ocean temperature), scaled by the spatially-varying factor $F_{\text{melt}}$ to account for lower ice-shelf melt rates for the Ross and Ronne-Filchner ice shelves. This way the sensitivity to basal melt rather than the sensitivity to ocean temperature is tested.

## 4 Present-day Antarctic ice sheet simulation

### 4.1 Initialization

Model initialization to the modern Antarctic ice sheet geometry is based on the method by Pollard and DeConto (2012b) by optimizing basal sliding coefficients in an iterative fashion. This nudging scheme is combined with the Weertman-type power law equation for basal sliding but can be used in conjunction with the two types of grounding-line flux conditions. The model (with grounding lines and floating ice constrained as described above) is run forward in time, starting from modern observed





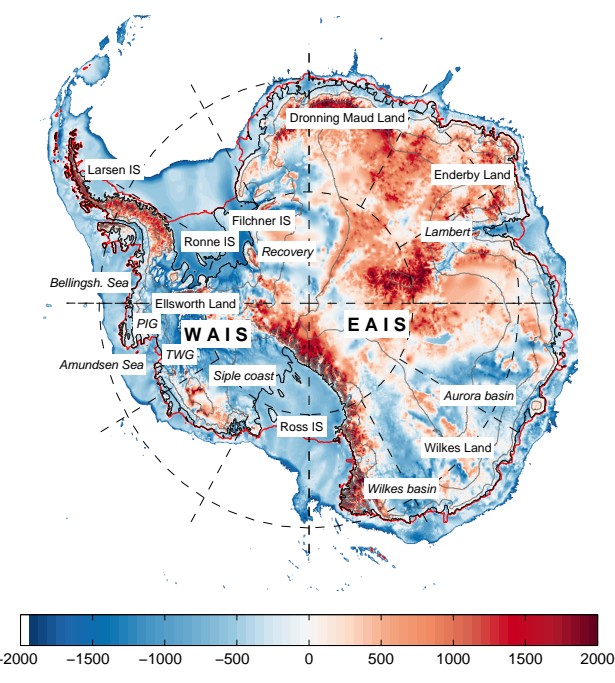

**Figure 3.** Bedrock topography (colour (m a.s.l.); Fretwell et al., 2013) and surface contours (grey; every 1000 m) of the Antarctic ice sheet, and ice sheet features mentioned in this paper. WAIS = West Antarctic ice sheet; EAIS = East Antarctic ice sheet; PIG = Pine Island Glacier; TWG = Thwaites Glacier; IS = ice shelf. Grounding lines are shown in black; ice shelf edges as a red line.

bed and ice surface elevations and further constrained by the observed climatology (surface mass balance and temperature). Full thermomechanical coupling and temperature evolution, isostatic bedrock adjustment as well as calving and sub-grid ice-shelf pinning is equally considered. Basal sliding coefficients $A_b(x,y)$ are initialized with a constant value ($A_b = 3 \times 10^{-9}$ m a$^{-1}$ Pa$^{-2}$) for the grounded ice sheet and a higher value ($A_b = 10^{-5}$ m a$^{-1}$ Pa$^{-2}$) underneath ice shelves and the ocean, to account

5    for slippery saturated marine sediments in case of re-grounding. At intervals of $\Delta t_{\mathrm{inv}}$ years, at each grid point with grounded ice, the local basal sliding coefficients $A_b(x,y)$ in Eq. (13) are adjusted by a multiplicative factor (Pollard and DeConto, 2012b):

$$A_b^\star = A_b \times 10^{\Delta z}, \tag{55}$$

where

10    $$\Delta z = \max\left[-1.5, \min\left(1.5, \frac{h_s - h_s^{\mathrm{obs}}}{h_s^{\mathrm{inv}}}\right)\right], \tag{56}$$





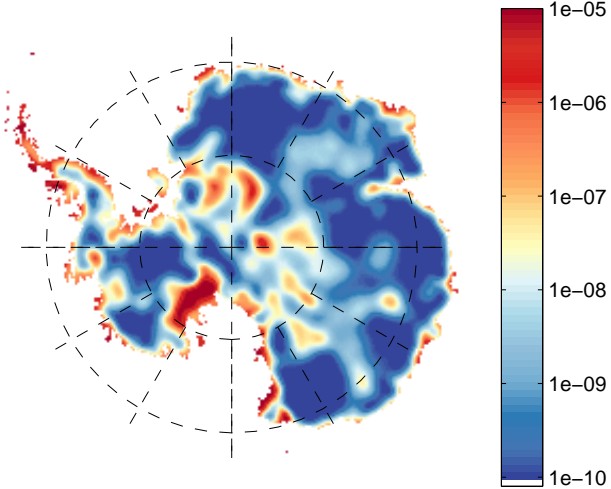

**Figure 4.** Optimized basal sliding coefficients $A_b^\star(x,y)$ after 100,000 years of integration with $h_{\mathrm{inv}} = 4000$ and $\Delta t_{\mathrm{inv}} = 500$ year.

and where $h_s^{\mathrm{obs}}$ is the observed ice thickness and $h_s^{\mathrm{inv}}$ is a scaling constant. During the inversion procedure, basal temperature is still allowed to influence sliding. Adjusted $A_b^\star(x,y)$ values are also not allowed to exceed $10^{-5}$ m a$^{-1}$ Pa$^{-2}$, representing the slipperiest deformable sediment. At the grounding line, observed surface velocities (Rignot et al., 2011) are used to define the buttressing factors at the grounding line in the grounding-line flux condition. Values for $A_b^\star$ are only updated when $r > 0$
in Eq. (14), so that they are kept unchanged when ice is frozen to the bedrock.

In addition to Pollard and DeConto (2012b) we also introduce a regularization term that essentially smooths high-frequency noise in the basal sliding coefficients by using a Savitsky-Golay filter of degree 3, with a span of 200 km (surrounding influence matrix). The influence matrix is thus made a function of horizontal distance instead of a fixed cell size. The advantage of such filter is that it keeps lower-frequency variability intact while removing high-frequency noise. This further improves the final fit
compared to the non-regularized case and it guarantees a smooth transition between the inland bedrock and the more slippery ocean beds under present-day ice shelves.

Optimized basal sliding coefficients (Fig. 4) for the Antarctic ice sheet on a spatial resolution of 25 km were obtained after a forward integration of 100,000 years with $h_{\mathrm{inv}} = 4000$ and $\Delta t_{\mathrm{inv}} = 500$ year. This results in a small difference between the observed and the steady-state modelled topographic surface (Fig. 5). For this run, the SIA model was preferred, as the velocity
constraint on the ice shelves does not require the SSA solution. Experiments with the hybrid model resulted in very similar results and a model drift after initialization comparable to the SIA model. The highest sliding coefficients are found in the marginal areas, especially in the Siple Coast sector, as well as under Pine Island and Thwaites Glaciers. Higher values are also encountered in the centre of the ice sheet, which is also obvious in other studies (Pollard and DeConto, 2012b; Bernales et al., 2016). These areas also show larger misfits (Fig. 5) and may be attributed to the poor knowledge of bedrock topography, so
that uncertainties are translated into a basal friction anomaly.





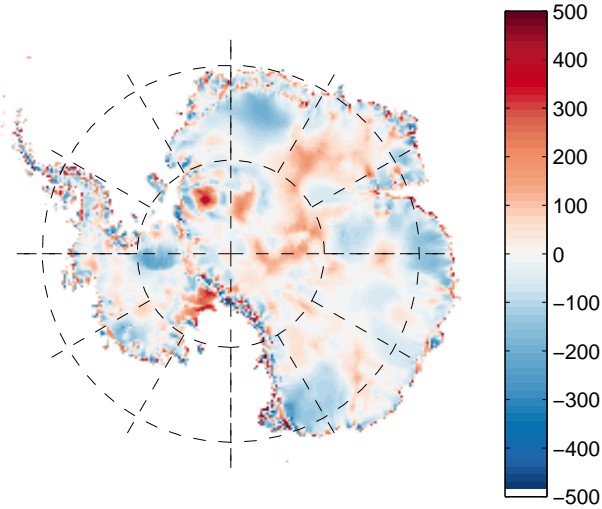

**Figure 5.** Difference between optimized and observed surface elevation after 100,000 years of integration with $h_{\mathrm{inv}} = 4000$ and $\Delta t_{\mathrm{inv}} = 500$ year.

Since the temperature field can be determined in steady-state, the time needed to reach a steady-state ice sheet is much shorter than in a conventional thermomechanically-coupled ice-sheet model. This allows for shorter integration times for convergence and updating intervals. The obtained patterns are in agreement with the results from Pollard and DeConto (2012a, b). The largest errors are found around the major mountain ranges (e.g., Transantarctic Mountains), since outlet glaciers protruding through these mountain ranges are not well represented on coarser grid cells. However, this fit has been improved by including bedrock variability in determining basal sliding coefficients $A_b'$ in Eq. (14) to allow for basal sliding of smaller outlet glaciers across mountain ranges.

### 4.2 Model validation

Modelled velocities form an independent check of the model performance, since the optimized basal sliding coefficients are obtained solely from the observed surface topography. The modelled flow field of the Antarctic ice sheet (Fig. 6) compares well to observations of surface velocities due to Rignot et al. (2011), such as the delineation of the different drainage basins and major ice streams discharging into the ice shelves. Some disagreement is found on glaciers discharging through the Transantarctic Mountains in the Ross ice shelf as well as glaciers near the Ellsworth Mountains discharging in the Ronne ice shelf. Those mismatches can be traced back by the difficulty in resolving those feature during the initialization process.

A direct comparison between the present-day velocity field (Rignot et al., 2011) and modelled velocities are shown in Figs. 7 and 8. The scatterplot (Fig. 7) shows a qualitatively good one-to-one fit for both the grounded ice sheet and the floating ice shelves. Quantitative error analysis shows a mean misfit of 19 m a$^{-1}$ with a standard deviation of 236 m a$^{-1}$ for the grounded





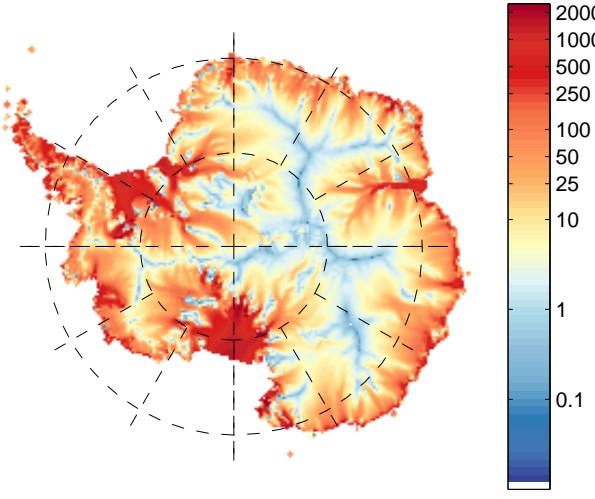

**Figure 6.** Modelled ice sheet surface velocities after optimization.

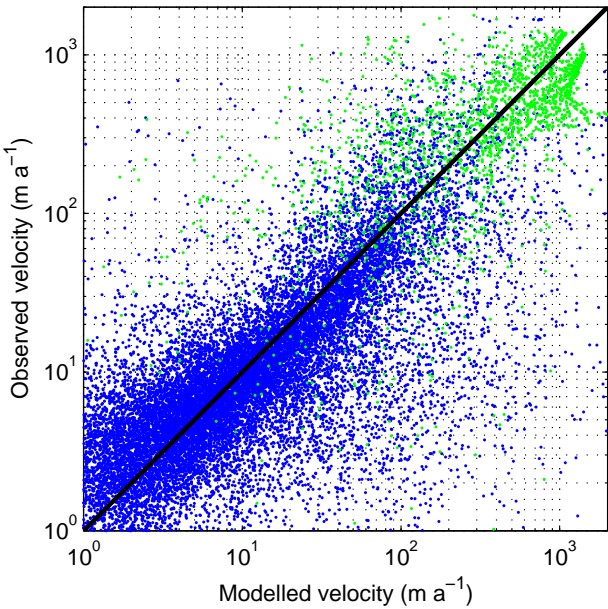

**Figure 7.** Point-by-point scatterplot of modelled and observed (Rignot et al., 2011) velocities. The mean difference from modelled to observed velocities for grounded points (blue) is 19 m a$^{-1}$ ($\sigma = 236$ m a$^{-1}$). For floating points (green) a larger difference of 57 m a$^{-1}$ ($\sigma = 549$ m a$^{-1}$) is obtained.





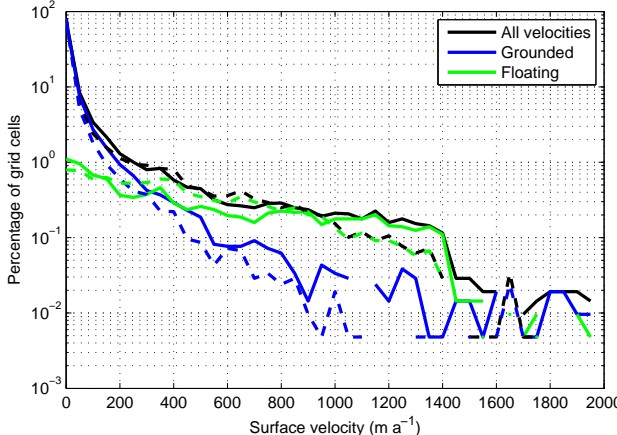

**Figure 8.** Histogram of velocity distribution of observed (dashed) and modelled (solid) velocities. Each of the bins contains a velocity range of 50 m a$^{-1}$.

ice flow, and a mean misfit of 57 m a$^{-1}$ with a standard deviation of 549 m a$^{-1}$ for the floating ice shelves. The histogram comparison (Fig. 8) demonstrates a good overall fit of observed and modelled velocity magnitudes. The modelled velocities are slightly higher than the observations, which can be attributed to the vertically-integrated nature of the model the approximation made in ice physics and thermomechanics. Nevertheless, the overall velocities (including ice shelves) map well with the observed ones and the result is in line with other model studies (e.g., Martin et al., 2011).

## 5 Sensitivity experiments

### 5.1 Sensitivity to ice-shelf de-buttressing

Ice shelves are the prime gatekeepers of Antarctic continental ice discharge. The breakup of the Larsen B ice shelf (Fig. 3) and the subsequent speed-up of outlet glaciers that previously discharged into the ice shelf witness this important instability mechanism (Scambos et al., 2000, 2004). In West Antarctica, observational evidence (Rignot et al., 2014) as well as modelling studies (Favier et al., 2014; Joughin et al., 2014; Seroussi et al., 2014) show that the reduction in buttressing of ice shelves in the Amundsen Sea embayment may lead to significant inland ice mass loss, and that unstoppable retreat of the grounding line of Thwaites Glacier may already be on its way (Joughin et al., 2014).

Since ice shelf buttressing is a key element in the stability of the Antarctic ice sheet, a useful experiment to understand underlying model buttressing physics is the sudden removal of all floating ice shelves, starting from the initialized model state, and to let the model evolve over time. Over this period ice shelves were not allowed to regrow, which is equivalent to a constant removal of all floating ice. This experiment is carried out for the two implemented grounding line physics, i.e., the flux condition according to Schoof (2007a) (SGL) and Tsai et al. (2015) (TGL), respectively. Both experiments result in a sudden ice-mass





**Figure 9.** Grounded ice sheet surface elevation (m a.s.l.), 500 years after sudden removal of all ice shelves (left), and grounding-line position in time according to the same experiment (right; colour scale is nonlinear and represents time (a)) for the SGL (top) and TGL (bottom) grounding-line flux conditions. SLR denotes the contribution to sea level rise after 500 years.





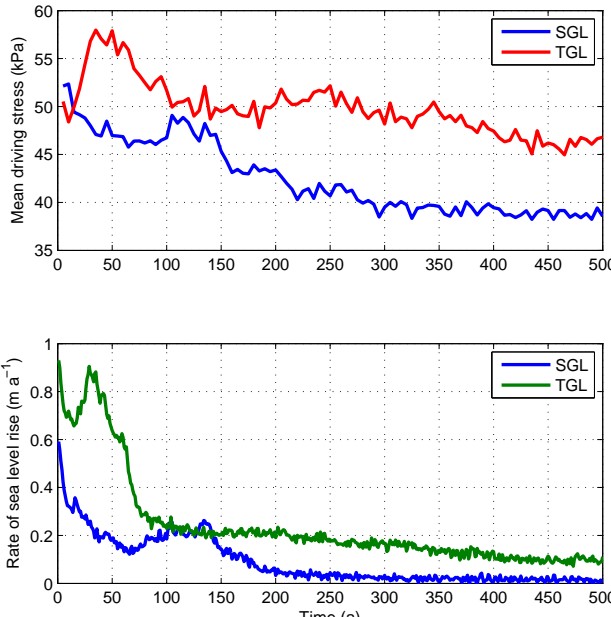

**Figure 10.** Evolution of the mean driving stress in the grounding zone – within 50 km upstream of the grounding line after sudden removal of all ice shelves (top) and corresponding ice mass loss – in terms of rate of sea-level rise (bottom) for the SGL and TGL experiments.

loss and grounding-line retreat, whereby the West Antarctic ice sheet collapses entirely in less than 200 years according to SGL and less than 100 years according to TGL, respectively (Figs. 9 and 10). For both experiments, grounding-line retreat starts in the marine sections discharging in the Ronne and Ross ice shelves. For the SGL experiment, the retreat from Ellsworth Land leads to thinning in the inland sectors of the Pine Island basin, which after >50 years triggers grounding-line retreat from

5 Pine Island Glacier and subsequently Thwaites Glacier. Grounding-line retreat then spreads rapidly towards the Ross sector of the West Antarctic ice sheet, leading to a complete disintegration of the ice sheet within 150 years. However, for the TGL experiment, initial grounding-line retreat also occurs in the Amundsen Sea sector, whereby the retreat is much faster and the ice sheet collapses within less than 100 years. Another major difference between both experiments is that the total mass loss for TGL is three times as large compared to SGL, i.e., a contribution to sea-level rise of ∼12 m for TGL compared to ∼4.5 m

for SGL after 500 years. The extra mass loss is essentially located in the East Antarctic ice sheet, i.e., Wilkes and Aurora basins (Wilkes Land; Fig. 3), both losing substantial amounts of ice. Despite the presence of a sill at the outlet of Wilkes subglacial basin, grounding-line retreat occurs without invoking any other physical mechanism than the flux condition at the grounding line in combination with complete ice shelf collapse. These results contrast with Mengel and Levermann (2014) who require the removal of a specific coastal ice volume equivalent to 80 mm of sea level rise in order to provoke an unstable grounding-line

retreat within Wilkes basin.





The higher TGL grounding-line sensitivity must be sought in its underlying physics: at the grounding line the basal shear stress vanishes in a smooth way to reach zero exactly at the grounding line. As shown by Tsai et al. (2015), this is not the case for the SGL algorithm, where a sharp contrast between the inland non-zero basal shear stress and the ocean exists. This boundary becomes smoother with larger sliding velocities, leading to a larger transition zone (Pattyn et al., 2006; Gladstone et al., 2012; Feldmann et al., 2014), but the transition jump does not vanish. For both cases (SGL and TGL), removal of ice shelves leads to an increase in driving stresses at the grounding line, mainly due to steeper surface slopes. As shown in Fig. 10, where the mean driving stress in the region within 50 km upstream from the grounding line is plotted in time, driving stresses increase when sudden mass loss is provoked. An increase in driving stress is therefore coincident with the collapse of the West Antarctic ice sheet (note the sudden increase in the rate of sea level rise; Fig. 10). While this is valid for both flux conditions, TGL is characterized by higher driving stresses throughout, hence a more important ice discharge, which facilitates unstable grounding-line retreat. This higher sensitivity is also demonstrated in the modified MISMIP experiments (Appendix C).

## 5.2 Sensitivity to sub-shelf melt

Antarctic ice sheet sensitivity to sub-shelf melting is investigated with a multi-parameter/multi-resolution forcing ensemble over a period of 500 years. A few experiments were also run over 5000 years. Atmospheric forcing includes changes in background temperature $\Delta T$, ranging from 0 to +8.5°C, affecting both surface temperature, Eq. (50), and surface mass balance, Eq. (51), through the mass balance–elevation feedback. Surface melt is calculated with the PDD model, Eq. (52). Ocean forcing is based on constant forcing values of sub-shelf melting $\Delta M$, ranging from 0 to 50 m a$^{-1}$ underneath the freely floating ice shelves surrounding the Antarctic ice sheet, and between 0 and 6.25 m a$^{-1}$ for the Ronne-Filchner and Ross ice shelves (factor 8 less compared to the freely-floating ice shelves). Melting is not allowed to be spread out across the grounded part of the ice sheet near the grounding line as is done in some models (Feldmann et al., 2014; Golledge et al., 2015). All forcings are applied as a sudden change in temperature/melt rate starting from the initialized model. A background run (without applying the forcing anomaly) is also performed to determine the model drift on the different time scales. The experiments are run for different combinations of sudden changes in background temperature/basal melting rate underneath the ice shelves on a grid size of $\Delta = 25$ km (as well as on a $\Delta = 40$ km grid to test grid-size dependence). A few runs are performed on a grid size of $\Delta = 16$ km for comparison. This gives a total of 40 forcing experiments for the $\Delta = 25, 40$ km grid spacings, and a further 10 experiments (considering only sub-shelf melt forcing) over a time span of 5000 years.

Sea-level contribution according to the forcing experiments and rate of change of sea level for the $\Delta = 25, 40$ km spatial resolutions are shown in Fig. 11. These are determined from the change in ice volume above floatation, hence do not represent the total grounded ice mass loss (Bindschadler et al., 2013; Nowicki et al., 2013). Sea-level change according to the forcings ranges between -0.5 and 6.5 m after 500 years. Sea level rise increases with increasing sub-shelf melt rates and slightly decreases with increasing atmospheric temperature forcing. The latter is due to the increased precipitation rates in a warmer climate, leading to an increase in grounded ice mass. The different curves in Fig. 11 are clustered according to sub-shelf melt rate, which is the most decisive process governing mass loss. Atmospheric forcing, however, has only a limited effect, probably because the time scale considered (500 years) is too short to relax the ice sheet to the imposed temperature and



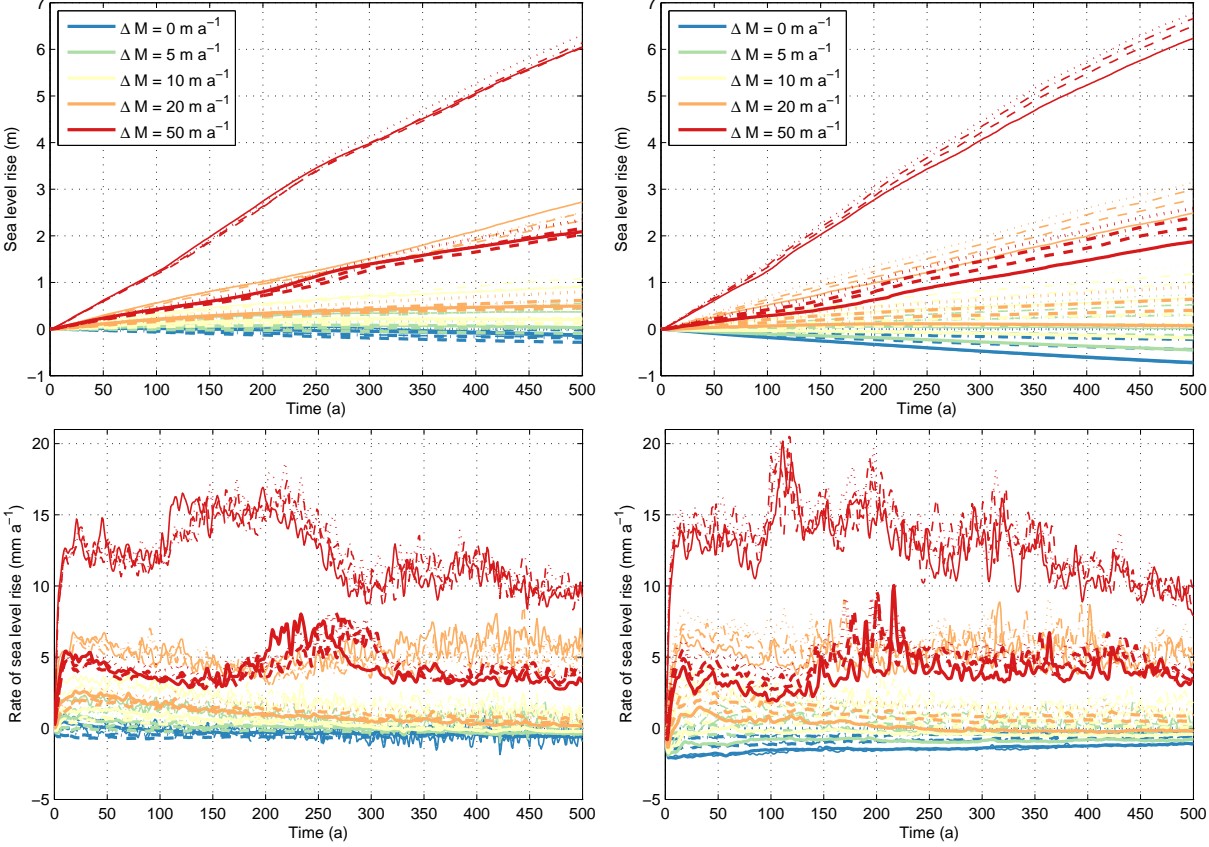

**Figure 11.** Evolution of sea-level contribution (top) and rate of sea-level rise (bottom) as a function of basal melting underneath ice shelves and background temperature change for the 25 km (left) and 40 km (right) spatial resolutions. Temperature forcing is as follows: 0°C (dotted), 2.2°C (dash-dot), 4.5°C (dashed), and 8.5°C (solid line). The thick lines correspond to the SGL grounding-line flux, while the thin lines correspond to the TGL flux.

precipitation changes. Model drift (zero forcing anomaly) is between 60 and 75 cm of sea level lowering over a period of 500 years, or 0.2–0.3% of the total Antarctic ice sheet volume per Century. This is comparable to other Antarctic model studies (e.g., Nowicki et al., 2013) and shows that the initialization is rather stable and close to steady-state.

5    The major discrepancy in sea-level response is with respect to the treatment of grounding-line fluxes. The TGL flux condition systematically leads to significant higher mass losses, making grounding-line migration a more sensitive process as already shown in Sect. 5.1. The higher sensitivity leads to a rate of change in sea level of up to 20 mm a$^{-1}$. These high values correspond to periods when the marine ice sheet runs into a major instability (MISI). Note, however, that such rates are still significantly lower than those obtained during the ice-shelf removal experiment (up to 1 m a$^{-1}$; Fig. 10). For the SGL flux condition, these values are half as much, and major MISIs occur generally at a later stage during the model run. Compared




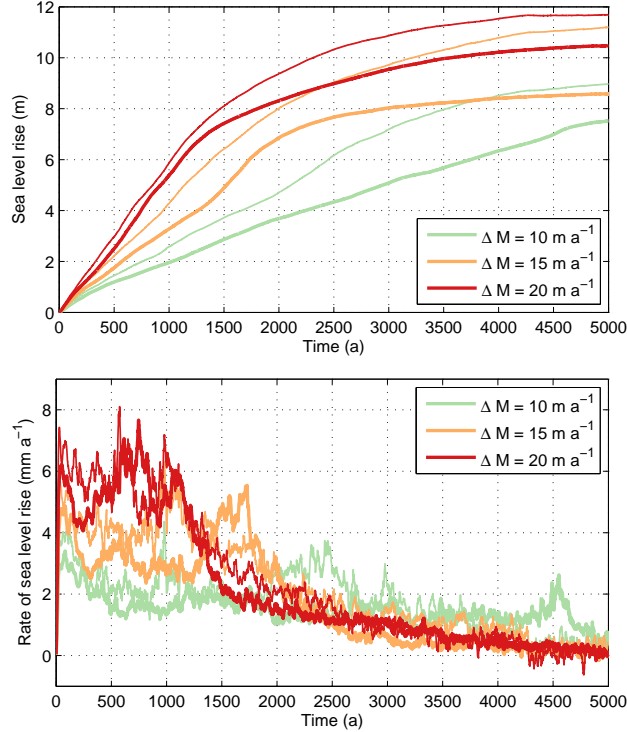

**Figure 12.** Evolution of sea-level contribution (top) and rate of sea-level rise (bottom) as a function of basal melting underneath ice shelves for 25 km (thick lines) and 40 km (thin lines) spatial resolutions and the TGL flux condition over 5000 year. Atmospheric forcing is not considered.

to other studies (Golledge et al., 2015; Ritz et al., 2015; DeConto and Pollard, 2016), the TGL flux conditions puts sea-level contributions at the high end of the spectrum and is comparable to the more 'aggressive' grounding-line migration setup in Golledge et al. (2015).

Only the higher melt-rate scenarios (20–50 m a$^{-1}$) produce significant MISIs over this time period. They first occur in the West Antarctic ice sheet (WAIS), starting from either Pine Island or Thwaites Glacier, progressing inland. Other MISI-prone areas are the Bellingshausen Sea (WAIS) and Wilkes basin (East Antarctic ice sheet – EAIS). Contrary to the de-buttressing experiment in Sect. 5.1, MISIs are not initially triggered in the Siple Coast area, nor through Ellsworth Land. This is probably due to the lower imposed melt rates, so that both Ronne and Ross ice shelves remain buttressed for a longer period of time. However, over longer time spans (5000 year), MISIs in the West Antarctic ice sheet seem to occur for lower melt rates.

Sea-level change over millennial time scales (5000 years) is investigated for the TGL flux condition without atmospheric forcing (Fig. 12). Here, only melt rates up to 20 m a$^{-1}$ were considered, so that rate of sea-level change are lower compared to the previous experiment. Most MISIs occur in the first 1000 years (for the highest melt rates) and within 2000 years for the 15 m a$^{-1}$ rates. MISIs occur at later periods (and are also less pronounced) for melt rates of 10 m a$^{-1}$. However, after 5000




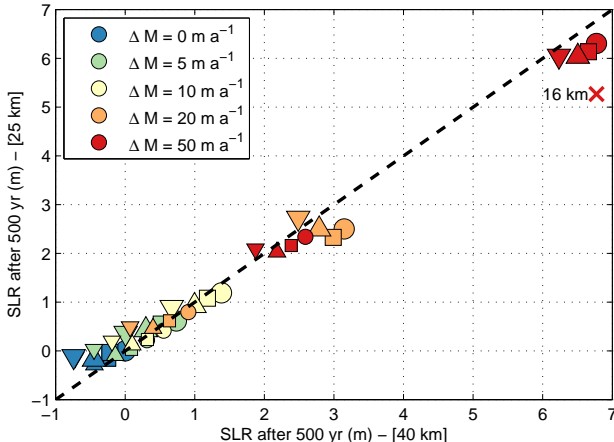

**Figure 13.** Comparison of sea-level contribution after 500 years as a function of model resolution. Colours denote sub-shelf melt rates; shapes represent background temperature forcing: $0°C$ (circles), $2.2°C$ (squares), $4.5°C$ (triangles), $8.5°C$ (inverted triangles). Small markers denote the SGL grounding-line flux condition, while large markers the TGL condition. Results for the $50 \text{ m a}^{-1}$ melt rate/no forcing anomaly/Tsai-flux experiment at a spatial resolution of 16 km is denoted by a red cross (and compared to the 40 km results).

years, sea level contribution is comprised between 7.5 and 10.5 m for all scenarios. They all represent a major destabilisation of the West Antarctic ice sheet. The higher melt scenarios also present significant contributions of the East Antarctic ice sheet (primarily Wilkes and Aurora basins).

The effect of spatial resolution on model result is summarized in Fig. 13 in addition to the data presented in Figs. 11 and 12. Coarser resolutions (40 km) give comparable results to the 25 km grid spacing with an almost one-to-one fit of sea-level contribution after 500 years between both resolutions. Both flux conditions follow this same fit. Larger deviations are observed over longer time spans of several millennia, but the timing of the major MISIs is comparable between grid resolutions (Fig. 12). The main reason for this relatively good fit must be sought in the grounding line flux conditions (SGL and TGL) that make the model resolution-independent. Models that are not based on such heuristics have to resolve grounding line migration at sub-kilometre resolutions (Pattyn et al., 2013; Pattyn and Durand, 2013).

However, it is expected that at high spatial resolutions, grounding-line retreat is influenced by bedrock irregularities as well as the presence of ice-shelf pinning points that are not always properly resolved at coarser resolutions. The parametrization of sub-grid processes, such as basal sliding in mountainous areas and sub-shelf pinning at sub-grid level, have to some extent reduced this dependency in the model. Despite these improvements, higher spatial resolutions (16 km, for instance), systematically lead to a smaller mass loss for a given forcing compared to the coarser resolutions (Fig. 13). Here, the effect of bedrock highs starts to play a role in delaying grounding line retreat (Durand et al., 2011). However, the overall contribution to sea level on longer time scales remains comparable to the results at lower spatial resolutions.



## 6 Discussion

In terms of model complexity, the f.ETISh model is comparable to the Pollard and DeConto (2012a) model. The major difference lies in a number of simplifications that makes the f.ETISh model two-dimensional. This is obtained by approximating the temperature calculation in a semi-analytical fashion (Pattyn, 2010) and by coupling a mean ice-column temperature to the velocity field via the commonly-used Arrhenius relationship (Cuffey and Paterson, 2010). Another major difference pertains to the marine boundary, with a novel implementation of the grounding-line flux condition according to Tsai et al. (2015), based on a Coulomb friction law (TGL). This is compared to the traditional Weertman-type boundary condition (SGL) due to Schoof (2007a). Other approximations pertain to linearizations in the SSA equations and basal sliding laws. Finally, model initialization based on Pollard and DeConto (2012b) has been further extended with a regularization term that essentially smooths the basal friction field and makes the results independent of spatial resolution, since regularization is made a function of horizontal distance instead of number of grid cells. Moreover, the optimization does not involve an optimization of ice-shelf basal mass balance, since observed ice-shelf velocities are used to determine the amount of buttressing at the grounding line. The resulting initialization is characterized by a small drift once the grounding line is allowed to relax, of the order of 0.2–0.3% of the ice sheet volume in 100 years. Other marine elements such as hydro-fracture and cliff failure (Pollard et al., 2015; DeConto and Pollard, 2016) are not taken into account.

Given the major differences in approach with continental-scale ice-sheet models, such as AISM-VUB (Huybrechts, 1990, 2002), ANICE (de Boer et al., 2013), GRISLI (Ritz et al., 2015), ISSM (Larour et al., 2012), PISM (Bueler and Brown, 2009), PISM-PIK (Martin et al., 2011; Winkelmann et al., 2011; Golledge et al., 2015), PSU-ISM (Pollard and DeConto, 2012a), RIMBAY (Thoma et al., 2014), or SICOPOLIS (Sato and Greve, 2012), verification of the f.ETISh model requires a detailed comparison with existing benchmarks. These are generally based on results of the models cited above. The EISMINT benchmark (Huybrechts et al., 1996) shows that the ice-dynamical characteristics of f.ETISh are in very close agreement with the benchmark, despite a different numerical solution scheme. The basal temperature field is also in close agreement and allowed to better define thermal control parameters in the approximation. As is to be expected, the time evolution of the basal temperature field deviates to some extent from the benchmark, with smaller time lags compared to ice thickness variations. This needs to be taken into account when the model is used on longer time scales (glacial–interglacial simulations, for instance). However, as shown in the sensitivity experiments, the thermomechanical effect is not the dominant process in marine ice sheet behaviour, and may only be of importance when focusing on central/divide areas of ice sheets. The results of thermomechanical coupling of ice sheet flow is despite the approximations also in good agreement with the EISMINT benchmark (Payne et al., 2000). Although deviations from the mean are larger compared to the previous benchmark, the range of uncertainty between the different participating models on which the benchmark is based, is also much larger.

An important experiment for marine ice sheet models is a test of steady-state grounding-line positions in absence of buttressing (Pattyn et al., 2012). Boundary layer theory indeed predicts that unique grounding line positions exist on a downward sloping bed, while no stable solutions are found on reversed bed slopes (Schoof, 2007a), unless buttressing is significant (Gudmundsson et al., 2012). While the experiments are designed for flowline models, they can be extended to two dimensions





to evaluate the behaviour in a qualitative way. Here, the f.ETISh model successfully passes the test independent of model resolution, as grounding-line migration is governed through a heuristic based on the above-mentioned boundary layer theory (Pollard and DeConto, 2009, 2012a) and is extended with a heuristic based on Tsai et al. (2015), that qualitatively gives the same results.

The main advantage of using a grounding-line flux parametrization based on a heuristic rule is that the model can be run at lower spatial resolutions, which is confirmed by the f.ETISh model experiments in Sect. 5.2. Solving the force balance around the grounding line requires to resolve membrane stresses at both sides of the grounding line with sufficient detail Schoof (2007a), which requires the use of sub-kilometre grid sizes (Pattyn et al., 2012), unless sub-grid grounding-line parametrizations are used that generally allow for grid sizes of ≈10 km (Feldmann et al., 2014). The main disadvantage of the heuristic

rule is that its parametrization is derived from a steady-state solution based on the SSA model. It can therefore be questioned whether the formulation still holds for transients. It also overrules the hybrid model at this particular location. Nevertheless, comparison with high-resolution SSA and hybrid models show that while differences in transient response exist, results are in overall agreement with the other models (Pattyn and Durand, 2013).

A major finding in this paper is the increased sensitivity of the grounding line based on a Coulomb friction law (Tsai et al.,

2015), compared to a power-law sliding condition at the grounding line. Power-law sliding mechanisms near grounding lines have been extensively discussed, since they lead to sudden jumps in basal drag at the grounding line, especially at relatively low sliding speeds (such as in the MISMIP and MISMIP3d experiments Pattyn et al., 2012, 2013). However, sliding velocities in the Antarctic experiments are not preconditioned by a specific sliding coefficient at the grounding line, but determined from the optimization procedure. Therefore, the type of boundary is controlled by the model physics itself. The Coulomb

friction condition at the grounding line is consistent with observations, as the ice-sheet profiles 'taper off' towards a flattening upper surface, contrary to the power-law case, and basal stresses vanish at the grounding line (Tsai et al., 2015). Moreover, the grounding-line ice flux according to Coulomb friction also depends more strongly on floatation ice thickness, implying higher sensitivity to atmospheric and ocean forcing. Furthermore, grounding is facilitated in shallower water compared to the power-law case, so that smaller perturbations may push the grounding line more easily into regions with a retrograde slope,

provoking a grounding-line instability (Tsai et al., 2015). As a result of the higher sensitivity, Antarctic sea-level contribution to a given perturbation is also more than twice as high and rates of sea-level change three times as fast compared to a power-law sliding case.

Direct comparison with other recent study on Antarctic ice mass loss is less evident, as most comprehensive studies follow so-called RCPs (Representative Concentration Pathways) that force atmosphere-ocean models. Direct comparison with the

SeaRISE experiments (Bindschadler et al., 2013; Nowicki et al., 2013) is also hampered due to the lower melt rates applied to the Ross and Ronne-Filchner ice shelves. This differentiation was deliberately chosen, as the de-buttressing experiments show that the highest buttressing stems from those large ice shelves. However, their grounding lines are also farthest from the continental shelf break, hampering the intrusion of warmer waters compared to the smaller ice shelves that are closer to the edge.





However, considering the f.ETISh model with the SGL condition comparable to the PSU-ISM model (Pollard and DeConto, 2009, 2012a), some comparison on sensitivity can be made. For the SeaRISE experiments, the PSU-ISM model predicts a sea-level contribution after 500 years according to a $2 \times$ A1B scenario (without sub-shelf melting) of $\sim$0.45 m, while the f.ETISh SGL model results in $\sim$0.4 m for similar forcing conditions. One has to note, however, that the initialization of both models is different (spinup versus optimization).

Golledge et al. (2015) presents a series of model runs over longer time spans (5000 years) with forcings that are kept constant for a prolonged period of time, which makes comparison possible. For a RCP8.5 scenario they obtain a sea level contribution of 5.2 m (9.3 m with sub-shelf melting spread out across the grounding line) and 8.6 (11.4) m for a RCP8.5 amplification scenario. Over the same period, f.ETISh covers the range of 8–12 m for moderate melt rates between 10 and 20 m a$^{-1}$. This shows that even the SGL model is more sensitive than the standard PISM model, but less sensitive when melting is allowed to be spread out across the grounding line (so-called 'aggressive' grounding line in PISM). The TGL model, on the other hand, systematically produces a higher contribution to sea level.

However, the TGL model is less sensitive than the PSU-ISM model including cliff failure and hydrofracturing (DeConto and Pollard, 2016). These processes potentially lead to a sea level contribution of 12-13 m after 500 years under a RCP8.5 scenario forced by atmosphere/ocean models. This result corresponds remarkably well with the results of the f.ETISh TGL model under complete de-buttressing (without ice-shelf growth), with complete collapse of the West Antarctic ice sheet and major ice loss in the Wilkes and Aurora basins (Fig. 9).

Finally, computational time of f.ETISh largely depends on the spatial resolution, which also governs time steps needed under the CFL condition. A hybrid-model 5000 year run with a grid size of 40 km and a time step of 0.2 year takes approximately 10,000 CPU seconds on a single AMD Opteron 2378 2.4 GHz core of the Hydra cluster (VUB-ULB) and 20,000 CPU seconds for a 500 year run with a grid size of 16 km and time step of 0.02 years on a multicore. Future developments will focus on improving the numerical solution schemes in order to reduce the calculation time (larger time steps), especially at higher spatial resolutions.

## 7 Conclusions

I developed a new marine ice sheet model, based on common descriptions of ice physics (combined shallow-ice and shallow-shelf approximation) and novel implementation of parametrizations of thermodynamics and grounding line migration. The model has been extensively tested against existing benchmarks and has been shown to be scale-independent, with the exception of high spatial resolution where detailed bedrock variability may delay grounding-line response to atmospheric and ocean forcing. This makes the model extremely attractive to couple within Earth System models.

The model has been initialized to the present-day Antarctic ice sheet conditions in order to obtain initial steady-state conditions as close as possible to the observed ice sheet. Independent validation has been obtained through comparison with observed surface velocities that are not utilised during the optimization phase.





Two forcing experiments over a period of 500 years are carried out, one during which all floating ice shelves are removed, and one during which sudden atmospheric and oceanic forcing is applied. Both experiments show a very high sensitivity to grounding-line conditions, as Coulomb friction in the grounding-line transition zone leads to significantly higher mass loss in both West and East Antarctica, compared to commonly-used power-law sliding laws (such as Weertman-type). For the

ice-shelf removal experiment this leads to 4.5 m and 12.2 m sea-level rise for the power-law basal sliding and Coulomb friction conditions at the grounding line, respectively. This high-end response is of the same order of magnitude as obtained by DeConto and Pollard (2016) using ice-shelf de-buttressing caused by hydrofracture and cliff failure.

The atmospheric/oceanic forcing experiments clearly show the dominance of ocean forcing in sea-level response, where significant MISIs (Marine Ice Sheet Instabilities) occur under relatively mild sub-shelf melt scenarios over centennial time

scales (500 years). Such MISIs seem to occur even for melt rates within the range of 1.25–10 m a$^{-1}$ over millennial time scales (5000 years).

## 8   Data availability

All datasets used in this paper are publicly available, such as Bedmap2 (Fretwell et al., 2013) and geothermal heat flow data (Purucker, 2013). Results of the RACMO2 model were kindly provided by Melchior Van Wessem.

*Acknowledgements.* I should like to thank Lionel Favier and Heiko Goelzer for the numerous discussions that helped in developing and improving the f.ETISh model and their helpful comments on an earlier version of the manuscript. I am also indebted to my 'guinea pigs' Thomas Bogaert, Violaine Coulon and Sainan Sun for revealing a few coding errors as well as for their patience while struggling with initial and non-optimized versions of the model.

## Appendix A: EISMINT I benchmark

### 20  A1   Fixed-margin experiment

The EISMINT I benchmark is the first series of ice-sheet model intercomparisons aiming at benchmarking large-scale ice sheet models under idealized and controlled conditions (Huybrechts et al., 1996). The first (fixed margin) experiment considers a square grid of 1500 × 1500 km with a flat bed at zero elevation. Grid spacing is taken as $\Delta = 50$ km leading to 31 × 31 regularly-spaced grid points. Starting from zero ice thickness, the model is forced with a constant surface mass balance of

0.3 m a$^{-1}$ and surface temperature according to $T_s = 239$ K $+(8 \times 10^{-8})d_{\mathrm{summit}}^3$, where $d_{\mathrm{summit}}$ is defined as $\max(|x - x_{\mathrm{summit}}|, |y - y_{\mathrm{summit}}|)$, expressed in km. Further boundary conditions for the model are zero ice thickness at the edges of the domain and a constant geothermal heat flux of $G = 0.042$ W m$^{-2}$. The ice temperature is not coupled to the ice flow field and a constant value for the flow parameter of $10^{16}$ Pa$^{-n}$ a$^{-1}$ is considered. The modelled ice sheet reaches a steady state in less than 25,000 years using a time step of 25 years, due to the fact that the temperature field is taken as steady-state (no relaxation

applied).




| Exp | Variable | Benchmark | f.ETISh |
|-----|----------|-----------|---------|
| FM | $h_{\mathrm{summit}}$ | 3419.90±1.70 | 3421.80 |
| | $q_{\mathrm{midpoint}}$ | 789.95±1.83 | 790.33 |
| | $T^b_{\mathrm{summit}}$ | -8.84±1.04 | -8.38 |
| MM | $h_{\mathrm{summit}}$ | 2997.5±7.4 | 2986.30 |
| | $q_{\mathrm{midpoint}}$ | 999.24±17.91 | 994.38 |
| | $T^b_{\mathrm{summit}}$ | -13.43±0.75 | -12.68 |

**Table A1.** Comparison of f.ETISh with the EISMINT I fixed (FM) and moving margin (MM) experiment benchmark based on an ensemble of 2–3 models (Huybrechts et al., 1996) for the steady-state experiment.

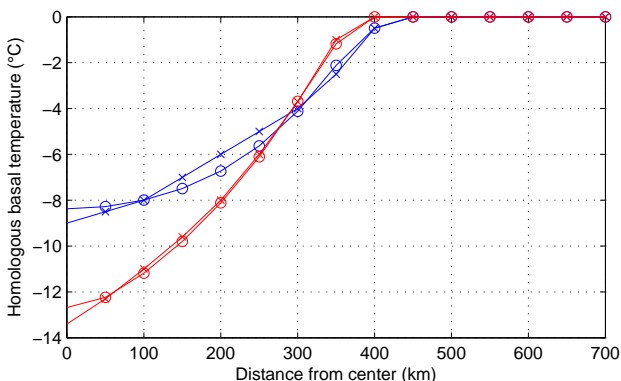

**Figure A1.** Homologous basal temperatures along the central line according to the EISMINT I experiment calculated with f.ETISh (circles) and according to the EISMINT I benchmark (crosses) for the fixed margin (blue) and moving margin (red) experiment.

The f.ETISh model is a 3d Type I model according to the classification scheme in EISMINT I, i.e., diffusion coefficients for the grounded ice sheet are calculated on a staggered Arakawa-B grid. Table A1 lists the comparison with data from other 3d Type I models. Both ice thickness and flux compare very well within error bounds of the sample range (limited to only 2–3 models in the EISMINT I benchmark, unfortunately). Also the basal temperature at the divide is within the limits given by the EISMINT I benchmark. The profile of the basal temperature in agreement with the benchmark (Fig. A1 has been obtained by setting $f_s = 0.25$ in Eq. (39). This way, strain heating at the base of the ice sheet is reduced to implicitly account for horizontal advection. Both processes are a function of the horizontal velocity, but act in opposing ways.

## A2 Moving margin experiment

The moving-margin experiment includes ice ablation, hence the presence of an equilibrium line on the ice sheet. This is obtained by defining the climatic conditions by $\dot{a} = \min\{0.5, h_s(R_{\mathrm{el}} - d_{\mathrm{summit}})\}$ and $T_s = 270 - 0.01h$, where $d_{\mathrm{summit}}$ is here defined as the radial distance from the centre (in km), and $s$ and $R_{\mathrm{el}}$ are $10^{-2}$ m a$^{-1}$ km$^{-1}$ and 450 km, respectively





(Huybrechts et al., 1996). The steady-state ice sheet according to this experiment does not reach the edge of the domain, but is circular in shape. Note that, contrary to the fixed margin experiment, surface temperature is a function of surface elevation and not of the geometrical characteristics of the domain. Surface mass balance, however, remains a function of the distance to the centre of the domain.

Basic characteristics of the experiment are listed in Table A1, and simulated values of ice thickness ($h_{\mathrm{summit}}$) and basal temperature at the divide ($T^b_{\mathrm{summit}}$), as well as ice flux between divide and margin are in good agreement with the benchmark. Also the basal temperature profile agrees well with the benchmark, for the same value of $f_s$ used in the fixed margin experiment.

### A3    Transient experiment

Temporal changes in ice thickness/volume and basal temperature are analysed with a forcing experiment, where the surface
temperature and mass balance perturbations are defined as follows (Huybrechts et al., 1996):

$$\Delta T = 10\sin\left(\frac{2\pi t}{T}\right), \tag{A1}$$

$$\Delta \dot{a} = 0.2\sin\left(\frac{2\pi t}{T}\right) \quad \text{for fixed margin}, \tag{A2}$$

$$\Delta R_{\mathrm{el}} = 100\sin\left(\frac{2\pi t}{T}\right) \quad \text{for moving margin}. \tag{A3}$$

The model run starts from the steady-state ice sheet obtained in the previous section and the forcing is applied for a period of
200 ka, with a periodicity of $T = 20$ and 40 ka, respectively. Results are depicted in Fig. A2 for the fixed margin and in Fig. A3 for the moving margin experiment. Table A2 lists the main characteristics of ice thickness and basal temperature amplitude variations, as well as ice thickness at the divide at the end of the experiment (200 ka).

All ice thickness changes according to the two forcing scenarios are in close agreement with the benchmark. However, amplitude and phase differences for the basal temperatures deviate. The phase response of basal temperatures at the ice divide is
much shorter for f.ETISh compared to the full thermodynamic calculation according to the benchmark, due to the approximation of the response time as a relaxation function. We did perform a series of sensitivity experiments (not shown) with varying tuning factors to the relaxation time (defined by the Peclet number), but this affected to a much larger extent the amplitude in response of the basal temperature signal rather than the shift in phase. All other parameters are within the bounds of the benchmark (Table A2).

## Appendix B:  EISMINT II benchmark

The EISMINT II benchmark (Payne et al., 2000) is based on the moving margin experiment of Huybrechts et al. (1996), but includes thermomechanical coupling of the ice flow to the temperature field. Contrary to the EISMINT I benchmark, inter-model differences are considerably larger, especially with respect to the area of the ice sheet that reaches pressure melting point at the base. The standard experiment consists of a flat bed of the same size as the EISMINT I benchmark, but with a



| Exp | Variable | Benchmark | f.ETISh |
|-----|----------|-----------|---------|
| FM 20ka | $h_{\mathrm{summit}}$ (200 ka) | 3264.8±5.6 | 3268.80 |
|  | $\Delta h_{\mathrm{summit}}$ | 563.0±3.7 | 565.94 |
|  | $\Delta T^{b}_{\mathrm{summit}}$ | 2.11±0.09 | **1.69** |
| FM 40ka | $h_{\mathrm{summit}}$ (200 ka) | 3341.7±3.9 | 3345.98 |
|  | $\Delta h_{\mathrm{summit}}$ | 619.0±3.2 | 621.60 |
|  | $\Delta T^{b}_{\mathrm{summit}}$ | 4.12±0.06 | **2.71** |
| MM 20ka | $h_{\mathrm{summit}}$ (200 ka) | 2813.5±2.0 | 2806.82 |
|  | $\Delta h_{\mathrm{summit}}$ | 528.6±11.3 | 533.88 |
|  | $\Delta T^{b}_{\mathrm{summit}}$ | 2.54±0.00 | **4.93** |
| MM 40ka | $h_{\mathrm{summit}}$ (200 ka) | 2872.5±6.8 | 2872.91 |
|  | $\Delta h_{\mathrm{summit}}$ | 591.4±4.6 | 595.27 |
|  | $\Delta T^{b}_{\mathrm{summit}}$ | 7.61±0.05 | 8.04 |

**Table A2.** Comparison of f.ETISh with the EISMINT I fixed (FM) and moving margin (MM) experiment benchmark based on an ensemble of 2–3 models (Huybrechts et al., 1996) for the forcing experiments with a sinusoidal signal of 20 and 40 ka, respectively. Bold values are those outside the range given by the benchmark results.

spatial resolution of 25 km, leading to $61 \times 61$ grid points. The basic experiment (A in Payne et al. (2000)) runs the ice sheet in equilibrium starting from zero ice thickness on the domain and with $u_b = 0$. The climatic conditions are defined as:

$$\dot{a} \quad = \quad \min\left\{\dot{a}_{\mathrm{max}}, s\left(R_{\mathrm{el}} - d_{\mathrm{summit}}\right)\right\} \tag{B1}$$

$$T_s \quad = \quad T_{\mathrm{min}} + s_T d_{\mathrm{summit}}, \tag{B2}$$

where $d_{\mathrm{summit}}$ is defined as in the moving margin experiment as the radial distance from the centre (in km), $s$ and $R_{\mathrm{el}}$ are taken as in the moving margin experiment ($10^{-2}$ m a$^{-1}$ km$^{-1}$ and 450 km, respectively), and $\dot{a}_{\mathrm{max}}$, $T_{\mathrm{min}}$ and $s_T$ are defined as 0.5 m a$^{-1}$, 238.15K, and $1.67 \times 10^{-2}$ K km$^{-1}$, respectively. Contrary to the moving margin experiment, climatic conditions are independent of ice sheet surface elevation, hence the mass-balance elevation feedback is excluded.

Four further experiments were carried out, i.e., experiment B, C, D and F (in Payne et al., 2000). They consist of a stepwise change in surface temperature, $T_{\mathrm{min}} = 243.15$K (B), a stepwise change in surface mass balance $\dot{a}_{\mathrm{max}} = 0.25$, $R_{\mathrm{el}} = 425$ km (C) and a stepwise shift in equilibrium-line altitude $R_{\mathrm{el}} = 425$ km. Experiments B, C and D start from the steady-state solution of A. Experiment F is similar to A, but starting with a value of $T_{\mathrm{min}} = 223.15$K (model run starting without ice).

Results for experiments A–D are summarized in Table A3. The majority of parameters are within the bounds of the benchmark, but major differences are related to the basal temperature at the divide. All experiments exhibit a radial pattern in basal temperatures that are at pressure melting point for the outer part of the ice sheet, with a cold spike in the center of the ice sheet. In all experiments, our temperature spike is less cold than the one given by the benchmark. However, despite this significant



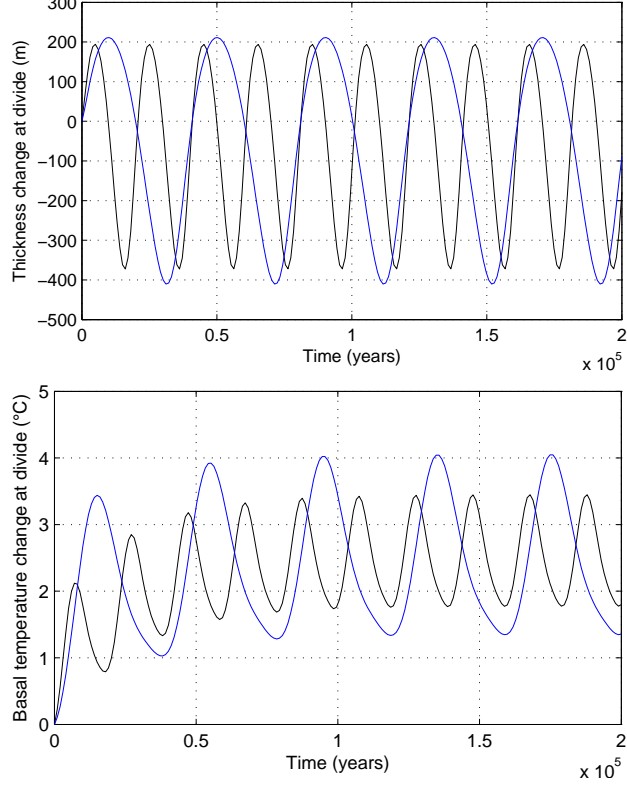

**Figure A2.** Ice thickness and basal temperature variations for the fixed margin experiment with a 20 ka (black) and a 40 ka (blue) forcing.

difference, the size of the basal area at pressure melting point is in accord with the benchmark. The main reason for this large difference is that temperatures in f.ETISh are calculated on a staggered Arakawa-B grid and not exactly at the ice divide, thereby always taking into account a given amount of strain heating due to the non-zero horizontal velocity. The difference is further exacerbated by the large horizontal temperature gradients for these experiments. As a result of the higher temperatures

5   under the ice divide, the simulated divide ice thickness is also lower than the one from the benchmark. Nevertheless, ice volume and area coverage are generally in accord.

The emblematic experiment F in Payne et al. (2000) displayed an irregular pattern in the basal temperatures of the benchmark for all participating models, leading to cold spikes reaching to the edge of the ice sheet. The pattern was shown to be model-dependent and further investigations traced its origin to an interaction between vertical advection (cooling down the base)

10   and strain heating (Hulton and Mineter, 2000). The pattern was found to be highly dependent on spatial grid resolution due to the lack of membrane streses in the shallow-ice approximation (Hindmarsh, 2006, 2009). Since f.ETISh does not account for vertical advection explicitly, the patterning is not produced by the model, even for a large range of surface temperature perturbations to provoke cooling at the base.





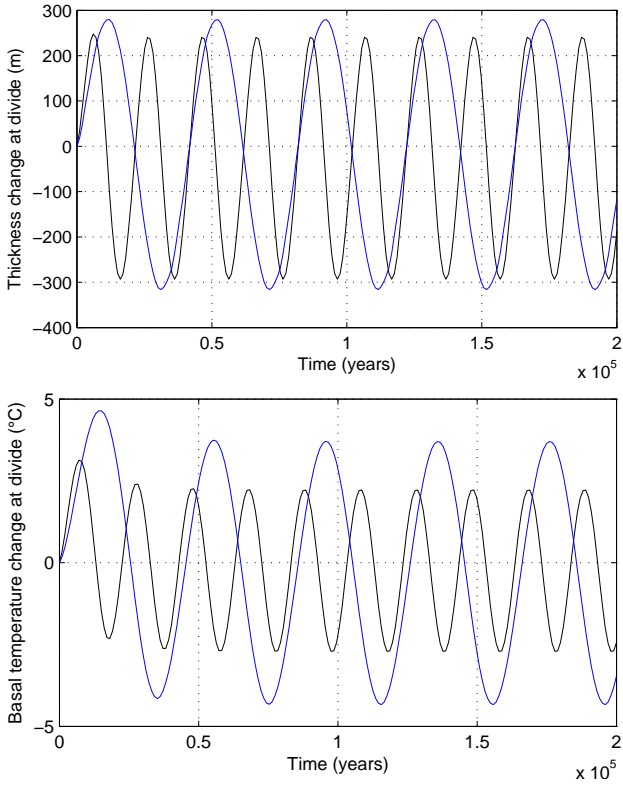

**Figure A3.** Ice thickness and basal temperature variations for the moving margin experiment with a 20 ka (black) and a 40 ka (blue) forcing.

## Appendix C: Modified MISMIP experiments

The capacity of an ice sheet model to cope with the marine boundary, and more specifically migration of the grounding line, is essential in Antarctic ice-sheet modelling. Since grounding-line dynamics were elucidated mathematically based on boundary layer theory (Schoof, 2007a, b, 2011), two intercomparison exercises were established. The first one tested grounding-line

5 migration and stability on downward sloping beds and instability on retrograde slopes for flow-line models (Pattyn et al., 2012), and the second tested the effect of buttressing for two- and three-dimensional ice-sheet models (Pattyn et al., 2013). Given that marine ice sheet instability is a crucial feedback process in marine ice sheet behaviour, we performed the flow-line experiments for a plan-view model setup. Experiments were carried out for both grounding-line flux conditions SGL and TGL. Ice shelves are included, but without exerting any buttressing strength, i.e. $\tau_{xx} = \tau_f$. The first experiment is an ice sheet on a

10 seaward-sloping bedrock, which in plan view results in a conic bed, defined by (Pattyn et al., 2012):

$$B = 720 - \frac{778.5}{750} d_{\text{summit}}, \tag{C1}$$





| Exp | Variable | Benchmark | f.ETISh |
|-----|----------|-----------|---------|
| A | Volume ($10^6$ km$^3$) | 2.128±0.145 | 2.007 |
|  | Area ($10^6$ km$^2$) | 1.034±0.086 | 1.041 |
|  | Melt fraction | 0.718±0.290 | 0.826 |
|  | $H_{summit}$ (m) | 3688.342±96.740 | **3354.515** |
|  | $T^b_{summit}$ (K) | -17.545±2.929 | **-6.500** |
| B | $\Delta$Volume (%) | -2.589±1.002 | -2.037 |
|  | $\Delta$Melt fraction (%) | 11.836±18.669 | 12.500 |
|  | $\Delta H_{summit}(\%)$ | -4.927±1.316 | -3.166 |
|  | $\Delta T^b_{summit}(K)$ | 4.623±0.518 | **2.323** |
| C | $\Delta$Volume (%) | -28.505±1.204 | -28.061 |
|  | $\Delta$Area (%) | -19.515±3.554 | -20.180 |
|  | $\Delta$Melt fraction (%) | -27.806±31.371 | -10.044 |
|  | $\Delta H_{summit}(\%)$ | -12.928±1.501 | -11.896 |
|  | $\Delta T^b_{summit}(K)$ | 3.707±0.615 | **-0.117** |
| D | $\Delta$Volume (%) | -12.085±1.236 | -12.565 |
|  | $\Delta$Area (%) | -9.489±3.260 | -10.090 |
|  | $\Delta$Melt fraction (%) | -1.613±5.745 | **7.666** |
|  | $\Delta H_{summit}(\%)$ | -2.181±0.532 | -2.445 |
|  | $\Delta T^b_{summit}(K)$ | -0.188±0.060 | -0.128 |

**Table A3.** Comparison of f.ETISh with the EISMINT II experiments (Payne et al., 2000).

where $d_{summit}$ (km) is the radial distance from the centre of the domain. The second experiment consists of an overdeepened section in the bedrock profile, hence the presence of a retrograde slope, defined by (Pattyn et al., 2012):

$$B = 729 - \frac{2184.8}{750^2}d^2_{summit} + \frac{1031.72}{750^4}d^4_{summit}$$
$$- \frac{151.72}{750^6}d^6_{summit} . \tag{C2}$$

5    The initial ice sheet is obtained for a constant value of the flow parameter $A$ of $10^{-16}$ Pa$^{-n}$ a$^{-1}$ and a constant surface mass balance of $\dot{a} = 0.3$ m a$^{-1}$. A grid-size spacing of $\Delta = 50$ km is employed. All other parameters are listed in Tables 1–3. Subsequently, the flow-rate parameter $A$ is altered to a new value to obtain a new steady state, where lower/higher values of $A$ leads to grounding-line advance/retreat, respectively. According to theory, a given set of boundary conditions leads to unique steady state grounding-line positions on a downward sloping bedrock, while the grounding line never reaches a steady-state

10    position on an upward-sloping bedrock, which is depicted in Fig. A4. For the overdeepened bed, this leads to hysteresis, i.e., multi-valued grounding-line positions and ice sheet profiles for the same set of boundary conditions (Figs. A4 and A5). The numerical error was estimated by determining the position of each grounding-line grid cell compared to its radial distance





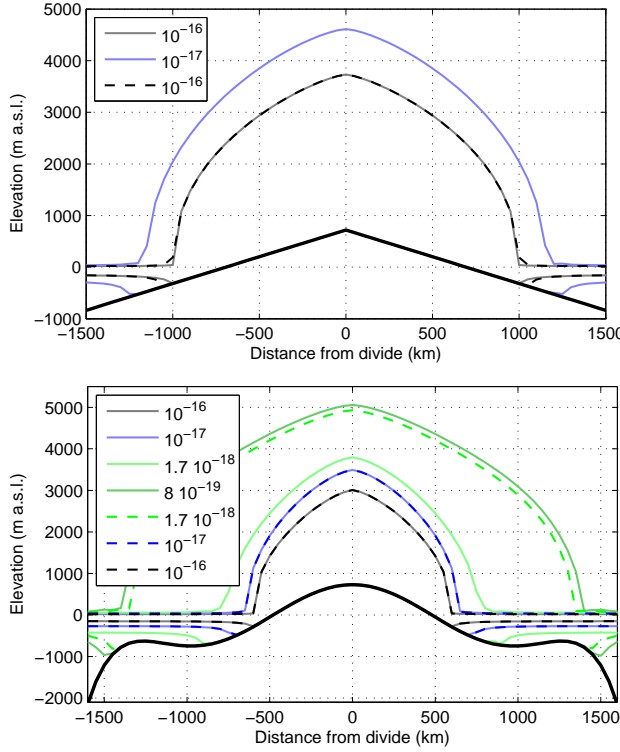

**Figure A4.** Steady-state ice-sheet/ice-shelf profiles corresponding to different values of the flow parameter $A$ ($\mathrm{Pa}^{-n}\,\mathrm{a}^{-1}$) along the center-line for the downward-sloping bedrock (upper panel) and the overdeepened bedrock case (lower panel) according to the advance (solid line) and retreat (dashed line) experiments and a grounding-line flux-condition according to Eq. (23).

from the centre of the ice sheet (both experiments results in radial ice caps). The mean position of the grounding line and the standard deviation corresponding to each steady-state are shown in Fig. A5. Interpolation of the exact position within a grid cell was not considered. All errors are smaller than the nominal grid size of 50 km. The lowest numerical error corresponds to the grounding-line treatment according to the power-law sliding law without the presence of ice shelves ($\sigma \sim 20$ km). Including

5  ice shelves makes the ice sheet more rapidly advance across the unstable section, since ice shelf thickness increases for lower values of $A$. Associated errors are also larger. Finally, the flux condition for Coulomb friction (Tsai et al., 2015) results in a generally smaller ice sheet, as the ice flux across the grounding line is higher than in the previous case. The ice sheet is also more sensitive to changes in $A$, i.e., small changes make the grounding line advance and retreat more rapidly. Associated errors are smaller for the no-shelf experiment, but significantly larger for the retreat experiment. Given the larger sensitivity,

10  the numerical solution is also less stable compared to the power-law flux condition SGL of Schoof (2007a).





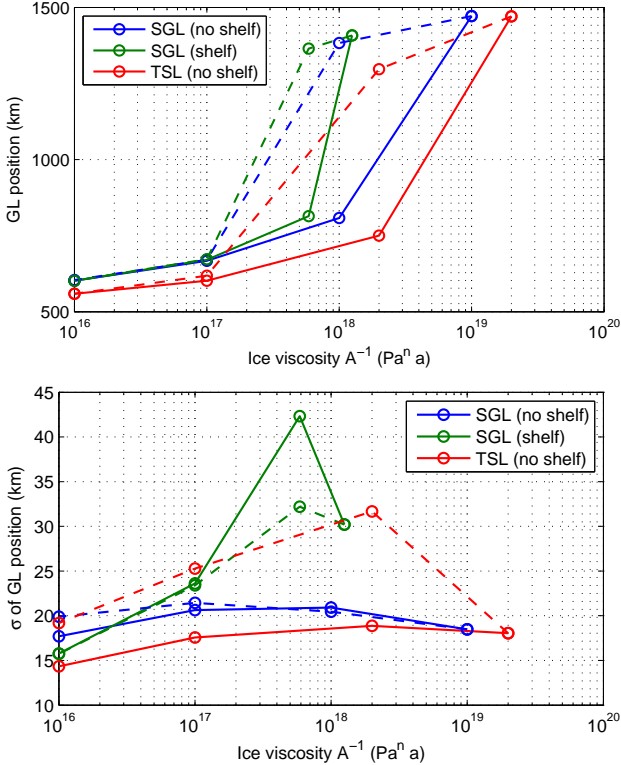

**Figure A5.** Median (upper panel) and standard deviation (lower panel) of steady-state grounding-line positions for a circular ice sheet as a function of the flow parameter $A$ $(Pa^{-n} a^{-1})$ for the overdeepened bedrock experiment according to different flux conditions at the grounding line and inclusion/exclusion of ice shelves. Solid lines represent advance and dashed lines represent retreat experiments.

**Appendix D: Ice-shelf velocity diagnostics**

EISMINT also provided ice-shelf test for diagnostic velocities of the Ross ice shelf (MacAyeal et al., 1996), which is repeated here, but compared to interferometrically-derived ice shelf velocities (Rignot et al., 2011). For this purpose, the model was run in diagnostic mode at a spatial resolution of 10 km with the Bedmap2 dataset (Fretwell et al., 2013). Ice flow velocities at

5 the grounding line are taken from Rignot et al. (2011) and are calculated for the shelf according to the linearised SSA model equations (3), (4) and (11). The ice-shelf velocity field was obtained with an adjustment flow-factor of $E_f = 0.05$ (Fig. A6). The magnitude of obtained velocities is also similar for other ice shelves with the same tuning factor (not shown).

The global Ross ice shelf velocity field is well reconstructed in the modelled result and matches the observed velocity magnitude. However, certain details of the flow field are missing, especially in relation to the outlet of Byrd Glacier, entering

10 the Ross ice shelf (Fig. A6). However, this flow feature is also missing in some velocity reconstructions from other models (MacAyeal et al., 1996) and is probably related to an underestimation of the ice flux across Byrd Glacier.





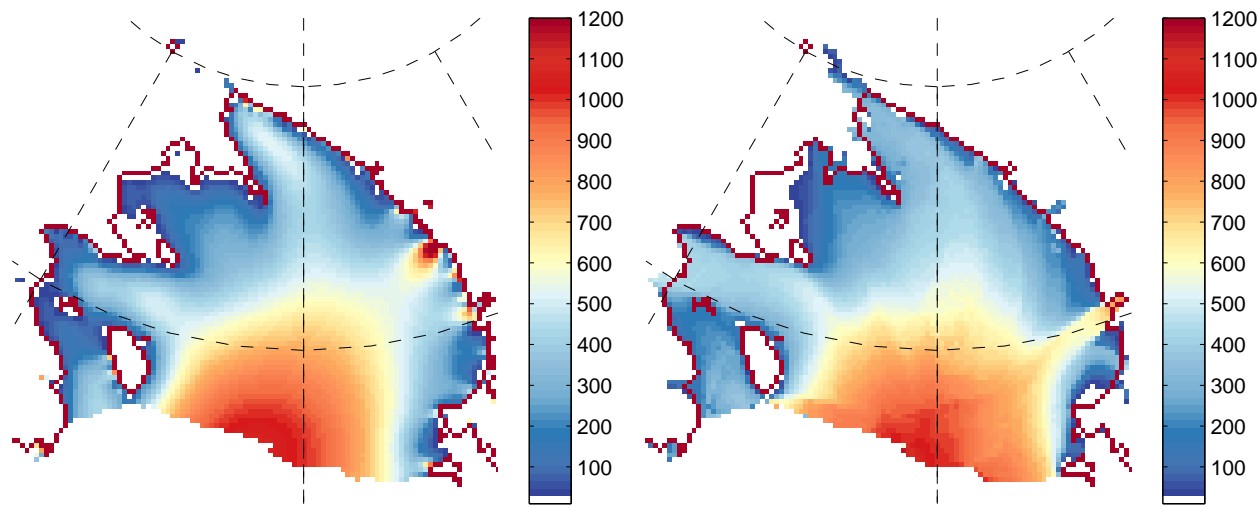

**Figure A6.** Comparison of modelled (left panel) and observed (right panel) ice shelf velocities (m a$^{-1}$) for the Ross ice shelf. Observed velocities are taken from Rignot et al. (2011) and resampled at a 10 km resolution.

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
