# Peer review of "Sea-level response to melting of Antarctic ice shelves on multi-centennial time scales with the fast Elementary Thermomechanical Ice Sheet model (f.ETISh v1.0)"

_The Cryosphere, 2017_

## Referee Comment (RC1) · Anonymous Referee #1 · 26 Feb 2017

**General comments:**

This paper presents a thorough and clear description of a new ice sheet model, akin to hybrid dynamics SIA/SSA models currently used for Antarctica, but with some reasonable and innovative simplifications so it is computationally fast. The model is implemented in MATLAB and will be a useful tool to engage students in teaching and workshop environments, as well as being capable for many research applications.

In this paper the model is thoroughly tested against established benchmarks (EISMINT, MISMIP) and validated vs. modern Antarctica. Sensitivity experiments of Antarctic retreat for simple warming perturbations are described. One important result is that much larger grounding line retreat is obtained with a Coulomb-friction based parameterization of grounding line flux, compared to that based on power-law sliding, but further testing may be desirable (see below).

Specific comments:

(1) The treatment of ice temperatures is based on classic vertical profile equilibrium solutions which allow for vertical ice velocity, and then time lagged with an e-folding relaxation towards these solutions at each grid point. The timescale of the e-folding lag is based reasonably on the local Peclet number (pg. 17, eq. 42). This is probably the most drastic simplification from other 3-D hybrid models, and neglects horizontal advection (which cools mid-level interiors as cold surface ice is advected downwards and outwards, and cools the cores of ice shelves supplied by flow across thick grounding lines. A fairly arbitrary compensation for this lack of cooling is attempted by reducing the strain heating (pg. 17, line 6). This simplified temperature treatment is evident in the benchmark intercomparisons in the Appendices, where basal temperature is the only field with poor results.

As a suggestion, perhaps basic horizontal temperature advection could be added to the model, ust by adding an additional term in Eq. (41): ... + u dT/x + v dT/dy with (u,v) given by (12) and T is the column mean temperature. That probably would not require much CPU or slowdown of the model.

Given this concern, I suggest that a map of the models basal temperatures for modern Antarctica be shown, and compared with existing model and data based maps (of which the author is a leader).

(2) It is puzzling why the inverse procedure for basal sliding coefficients (p. 23-24, Fig. 5) yields quite large errors in surface elevation ( $\sim$ 200 m) in some regions of the interior East Antarctic plateau. The inverse procedure should reduce them to 10's m (Pollard and DeConto, 2012b) (even if the bed elevations are in error, model or observed, cf.

TCD
pg. 24 line 19).

Perhaps these larger errors are due to regions of the bed erroneously being frozen. In frozen basal regions the inverse procedure cannot reduce the model's surface elevation errors. So this is an additional reason to request a basal-temperature map.

nb: "ice thickness", pg.24 line 1, should probably be "ice surface elevation".

(3) One important result is the greater grounding line retreat with TGL (Coulomb-friction based grounding line flux parameterization, Eq. 25), vs SGL (power-law sliding based, Eq. 23). All experiments shown use power-law sliding (Eq. 15) for the interior grounded ice, and none use Coulomb sliding (Eq. 21). My concern is that the combination of TGL with power-law interior sliding is not compatible, and the mismatch in the physics may lead to spurious behavior in grounding zone regions. (The discussion on pg. 13, lines 24-27 may be relevant).

To address this concern, I would request additional runs be made with Coulomb friction law (Eq. 21) and the TGL grounding line parameterization. This would ideally also involve re-doing the optimization spin-up for basal properties, which may still be feasible by changing phi (till friction angle) instead of A\_b in Eq. (55). Alternatively, the combined Eq. (22) could be used instead of (21).

(4) The use of driving stress instead of basal stress in the basal sliding law to avoid iterations (pg. 10, Eqs. 15,16) is one of the features used to speed up the model. But maybe the 20% of the ice sheet where driving stresses are not essentially balanced by basal stresses (p.10, lines 16-17) are in important regions such as ice streams. This concern could be addressed by one sensitivity test in which the approximation in Eq. (16) is not made (requiring expensive iteration).

(5) The subglacial water pressure p\_w in Eqs. (19) and (20), pg. 11, is assumed to depend on elevation minus sea level, which is a common step in many models. But it is hard to see how the subglacial water system can sense hydrostatic pressure from

TCD
the ocean at all, more than  $\sim$  100 or 200 km inland from the grounding line.

Technical points:

p.3, Fig. 1. I suggest indicating in the figure that sea level is at z=0, as seems to be required in Eqs. 18, 19 and 20. And z\_sl must = 0 (p.7, line 3). Alternatively, replace b throughout p.11 with b - z\_sl.

p.7, Eq (2). More correctly, v\_sia = v\_b + ...  $|tau_d|^n-1 tau_d$

p.9, line 7 et seq. To avoid confusion, say explicitly that tau\_f is the free-floating stress, used later in Eq. (24) as well as in (3),(4) via eta in (11).

p.11, line 3: Why might the friction angle phi be a function of bedrock elevation, physically?

p.12, lines 6-7. The sentence " However, expressed as a ... " is unclear to me.

p.13, lines 7-10. Here, it might be helpful to mention that a staggered grid (Arakawa C) grid is used as shown in Fig. 2.

p.14, Eq. (27). Say that this is only applicable for SIA advection.

p.14, lines 11-22: Say whether this 'maximum strain check' is applied everywhere, on ice shelves, or just at the grounding line.

p.17, line 24 and Eq.(42). Say that this is vertical advection (not horizontal).

p.18, line 13. Specify the value of E\_f used for ice shelves.

p. 19, lines 1 and 10. Say that the equilibrium bed topography and loads (b\_eq, h\_eq, h\_w\_eq) are taken from modern observed fields (Bedmap2), if that is the case.

p.19, line 18.Say that the local numerical integration is for Eq.(38) (I think).

p.19, line 24. Iterations are also eliminated due to the approximation of driving = basal stress in Eq. (16).
p.21, line 24. A simple one-valued PDD is ok for modern Antarctica with little surface melt. But the surface melt treatment will need improving (snow vs. ice, refreezing, etc.) to represent greatly increased surface melt around the Antarctic margins in warm future climates.

p.22, line 9. Say where ocean temperatures T\_oc are obtained from. Actually it seems that Eq. (53) and T\_oc are not used in any experiments here, for which the melt rate M is simply prescribed region by region (p.30, lines 17-19).

p.22, line 16. Eq. (53) produces higher melt rates closest to the grounding line not because it's quadratic, but because the freezing temperature T\_fo decreases with depth (noting h\_b in Eq. (54) is negative below sea level).

p.23, line 1. Perhaps change "further constrained by" to "driven by".

p.25, line 14. Change "back by" to "back to".

p.27, line 3. Perhaps change to "of the model and the approximations..."

p.28, Fig. 9 caption, 2nd line. Remove "(a)".

p.30, line 8. Perhaps change "provoked" to "applied".

p.30, line 10. \*Why\* are TGL runs characterized by higher driving stresses?

p.31, Fig. 11 caption, 2nd line. Say "Atmospheric temperature forcing is..."

p.34, line 22. Mention that the agreement with the benchmark(s) is shown in the Appendices.

p.34, lines 27-28. Change to "Despite the approximations, the results of thermomechanical coupling of ice sheet flow are also in good agreement with the EISMINT benchmark..."

pg.34, line 29: Perhaps change to "compared to the other benchmarks,"

pg.35, lines 7-8. Change to "requires membrane stresses at both sides of the ground-

TCD
ing line to be resolved with sufficient detail (Schoof 2007a),"

pg.35, line 10. Mention in parentheses that this rule is that on pg. 13, lines 9-10.

pg.35, lines 28-29. Change to "Direct comparison is not possible with recent studies of Antarctic ice mass loss that are forced by atmosphere-ocean models following so-called RCPs (Representative Concentration Pathways). Direct comparison with the SeaRISE...

pg.36, line 8. Perhaps describe in a few words what "RCP8.5 amplification" is.

pg.36, lines 28-29. Perhaps change to "with the exception of grounding-line zones with small-scale bedrock variability, where grounding-line response to atmospheric and oceanic forcing is sensitive to spatial resolution."

pg.37, line 8. The "dominance of ocean forcing" in this paper relies on the absence of physics such as hydrofracturing that occur due to large increases in surface melting around the margins. With RCP8.5 at least, there will be a huge increase in the latter within  $\sim$ 100 to 200 years, which could affect the ice sheet in unexpected ways.

Appendices, figure captions A2 to A5. It would help to specify the benchmark experiment (EISMINT I or II, MISMIP, etc) in each caption, especially if the figures appear on different pages than the relevant text.

TCD

---

## Referee Comment (RC2) · S. L. Cornford (Referee) · 3 Apr 2017

This manuscript presents the details and some experiments carried out with a new ice sheet model code (f.ETISH), which is designed to represent the major process in near future ice sheet dynamics (e.g changes in grounding line flux due to ice shelf thinning) well enough to be meaningful but not requiring fine spatial resolution and the attendant computational cost. The new model is similar in that respect to the model of Pollard and DeConto (sans cliff collapse), but makes some additional approximations for the sake of speed. Is this a useful new model? Perhaps. It could be very well

suited to long-term integrations, though that is equally true of Pollard and DeConto. It could be useful in large ensemble construction, where I think it makes a better job of representing the physics than (say) Ritz et al 2015. It does seem to contain new but not really well justified approximations in rheology and temperature structure, but given that these things are largely unknown and must be tuned anyway, that may not be very serious problem.

The paper is a bit rambling in parts (maybe this review is too), especially the model description and the discussion. Sometimes it includes expressions that are well known, then ignores them, e.g the effective viscosity is treated this way, and the discussion of Coulomb friction laws seem to be a bit extraneous too. I think it really does need a substantial review and edit.

**General Comments**

One message of the paper seems to be that treating the flux across the grounding line according to Tsai (TGL) rather than Schoof (SGL) dramatically increases the retreat rate. This could be because Tsai depends on a higher power of flotation thickness (more acceleration of retreat on retrograde slopes), but might to some extent be attributed to the addition of another free parameter (tan phi) . Given that the model is so quick, maybe some runs with larger $\tan(\phi)$? Looking at eq 18, for much of WAIS where bedrock elevation, is < -1 km, $\tan(\phi)$ is around $\tan(\phi_{min}) = 0.2$. What happens if $\tan(\phi) = \tan(\phi_{max}) \approx 0.5$ – presumably we see about half the rate of SLR? On the same note, TGL is double sided - flux increases more quickly with grounding line thickness, which would mean that the grounding line accelerates more readily in unstable configurations (retrograde slopes, without buttressing) but decelerates more readily in stable configurations (prograde slopes) - and the formula for tan(phi) should amplify this effect somewhat. The East Antarctic results in section 5 seem to differ from this,

with the introduction of TGL leading to retreat over prograde slopes (Totten, Wilkes Basin, Recovery Glacier a little upstream from the present day GL) where there was little in the results with SGL

I am suspicious of approximating the effective viscosity by assuming that the stress that enters it is that of an free floating 1HD shelf (eqs 8 and 9). Can this really be a good approximation in buttressed ice shelves like (e.g) PIG, Amery, Totten, where there are regions with little along flow stretching, but strong lateral strains ? To me, the example in appendix D is not especially convincing – cross flow gradients are too low, and the flow field too smooth. This might not be a big error in itself, since at higher melt rates all that will matter is the TGL / SGL with no buttressing, but a more convincing test is needed. Why not re-run a middle melt-rate experiment with the normal nonlinear rheology?

Given that there is a well known test - MISMIP+ - with published results that include both the Tsai friction rule and ice shelf buttressing - why not test f.ETISH against that?

**Specific comments**

Abstract

(and elsewhere) "The higher sensitivity [in the case of the Tsai 2] is attributed to higher driving stresses upstream from the grounding line." I'm not sure this makes sense – and I suggest it is at least partly the other way round. Because q(TGL) is larger than q(SGL), but both are only applied at the GL, dh/dx is going to be bigger at the GL for TGL with all else being equal. The same – plain Weertman - friction law is applied upstream.

Section 1

"The majority of these interactions demonstrate non-linear behaviour due to feedbacks, leading to self-amplifying ice mass change." -> "Some of these . . ."

"thicker ice grounded in deeper water would result in floatation, increased ice discharge, and further retreat within a positive feedback loop." -> thicker ice grounded in deeper water would result in increased ice discharge, and further retreat within a positive feedback loop.

". . .. based on boundary layer theory (...Ritz et al., 2015. . .)". I don't think the Ritz et al., 2015 GL is based on boundary layer theory, does it? But imposes retreat rates sampled from some sort of probability distribution.

Section 2.1

"The main advantage of SIA is that the velocity is completely determined from the local ice-sheet geometry." That might be called the main disadvantage too.

SSA+SIA : "a simple addition still guarantees a smooth transition" - why wouldn't it? SIA isn't smooth in the same way as SSA, but so long as the surface elevation is smooth, it will be. More to the point, is this a good approximation? How about at the ice shelf calving front, where grad(s) is large, there is no basal stick and SIA makes no sense? I don't think Schoof and Hindmarsh 2010 gives us a reason to think that SSA+SIA is any more sensible than plain SSA.

"Basal velocities in the hybrid model are defined through a friction power law, where" Basal traction, no? The velocity is related but depends also on viscous stress at least close to the GL in the SSA+SIA case.

2..1.7. The Coulomb friction law plays no part in the results, except for its involvement

with the Tsai flux. I suggest cutting this (longish) section 2.1.5 entirely and describing tan(phi) in 2.1.7

"where 'spy' is the number of seconds per year" Why switch units mid-expression?

Eq 25: What value does $tan(\phi)$ tend to take?

Section 2.2

eq 28: should be an inequality? $hdu/dx \leq AhT_f$ (ie the maximum stretching is in free shelves)

"Ritz et al. (2015) use a slightly different prescription, but sensitivity tests showed that the extra terms in the mass conservation equation can be safely dropped, rendering the maximum strain check therefore independent of velocity gradients." Which terms? The terms that have been dropped are $-udh/dx$ and $-vdh/dx$, both of which involve **thickness** gradients and are typically positive, so in fact

$$dh/dt \geq a - -M - -h(du/dx + dv/dy) \leq a - M - 2AhT_f$$

And you assume $dh/dx$ is neglibigle?

"However, to compensate for the absence of horizontal advection in the model, only a fraction $fs \approx 0.25$ of the total strain heating amount was added. This value is determined from the EISMINT benchmark experiments (Appendix A)." Should this value not depend at all on ice speed? Is ESIMINT a sufficient test of this quite different dynamics?

Section 2.3

OK, these expression come from others. But are they justified in any way.

Section 2.4

Eq 33. Has some horizontal advection - needed to eliminate $Tdw/dz$ and reduce vertical advection to $wdT/dz$, but neglects horizontal temperature variation? An even simpler solution might be possible if the conservative advection $d/dz(wT)$ was used and all horizontal transport neglected. Or did I miss something?

How is eq 41 based on the Peclet number? So $\tau_t$ is advection dominated when $Pe$ is large and diffusion dominated when $Pe \to 0$, but does the code actually compute some function of $Pe$?

Section 2.6

Why solve eq 46 with BiCGStab? What preconditioner is used? ILU(0)? UMFpack is MATLAB's default sparse solver, I think, and I guess the matrices are all small (coarse grid), so if there are large ice shelves (so A becomes poorly conditioned) this direct solver might be the better choice (or not)

Section 4.

"This further improves the final fit compared to the non-regularized case.." which is not normally the case with regularization - typically regularization results in worse (or no-better) fit to the observations for the sake of a smoother (or more plausible in some other sense) solution.

Section 5.1:

Sorry to bring this up, but Cornford et al 2016, Annals of Glaciology
https://doi.org/10.1017/aog.2016.13 does a rather similar experiment (all-Antarctic re-
sponse to sustained ice shelf removal), with a sub-km model, and the Weertman sliding
results could be compared.

Is the rate of SLR labelled incorrectly in fig 10?

Section 5.2

"Melting is not allowed to be spread out across the grounded part of the 20 ice sheet
near the grounding line as is done in some models (Feldmann et al., 2014; Golledge
et al., 2015)". Note that Feldmann and Golledge are not really trying to spread the
melting about, they are just applying some melt to finite area grid cells whose centers
are grounded but whose neighbours are floating, estimating a floating fraction by inter-
polating the thickness above flotation. This sounds pretty innocuous - even sensible -
in which context the sentence above sounds like the wrong choice. Of course we know
it is not the wrong choice, but maybe say something about why?

"[SLR] determined from the change in ice volume above floatation, hence do not rep-
resent the total grounded ice mass loss" Seems like an odd comment - how else would
it be computed ? It makes me wonder if the section 5.1 SLR is from total mass loss
(indeed the text of section 5.1 suggests that, "the total mass loss for TGL is three times
as large compared to SGL, i.e., a contribution to sea-level rise of 12 m ..."), when I
assumed it had been computed from VAF

Fig 11. Although the 'thick lines (SGL), thin lines (TGL)' plot works for the large delta
M, I can't make so well out what is going on at small delta M. how about thin lines with
a few symbols (say, circles, squares). Or drop the $\Delta M = 10$ m/a results?

[Figure]

Fig 12. To my mind, at least one more grid spacing (there are some runs 16km. right? ) to be able to say much about mesh dependence. You can't test convergence at all with just two, you need to show that results are getting closer to one another as $dx \rightarrow 0$

Section 6

"Another major difference pertains to the marine boundary, with a novel implementation of the grounding-line flux condition according to Tsai et al. (2015), based on a Coulomb friction law (TGL)" 'novel' seems a bit strong, given that Tsai derived the flux formula, and the implementation replaces a very similar formula (SGL) in an overall method to modify the Schoof flux to include buttressing due to Pollard.

p35 "unless sub-grid grounding-line parametrizations are used that generally allow for grid sizes of $\approx 10$ km (Feldmann et al., 2014). ". Personally I think this claim in Feldmann 2014 is not supported by the results, which are better with the sub-grid scheme, but still need dx   1 km. Why should we believe that results in one idealized problem should be widely true?

"Nevertheless,comparison with high-resolution SSA and hybrid models show that while differences in transient response exist, results are in overall agreement with the other models (Pattyn and Durand, 2013)." That really was not the message I took from Pattyn and Durand 2013, at least regarding the transient.

"as the ice-sheet profiles 'taper off' towards a flattening upper surface, contrary to the power-law case," - this happens to some extent in the power law case too, depending on the scale length for viscous stresses transmission.

"(so-called 'aggressive' grounding line in PISM)." Does Golledge really call it 'agressive' in that paper. I remember him saying it in a talk. Anyway, why not say what it is: a type of numerical error (aggression -> 0 as dx -> 0) rather than something that could be seen as physics.

---

## Author Comment (AC1) · 10 Apr 2017

**Response to the Interactive comment on "Sea-level response to melting of Antarctic ice shelves on multi-centennial time scales with the fast Elementary Thermomechanical Ice Sheet model (f.ETISh v1.0)" by Frank Pattyn**

Anonymous Referee 1

General comments:

[Figure]

This paper presents a thorough and clear description of a new ice sheet model, akin to hybrid dynamics SIA/SSA models currently used for Antarctica, but with some reasonable and innovative simplifications so it is computationally fast. The model is implemented in MATLAB and will be a useful tool to engage students in teaching and workshop environments, as well as being capable for many research applications.

In this paper the model is thoroughly tested against established benchmarks (EISMINT, MISMIP) and validated vs. modern Antarctica. Sensitivity experiments of Antarctic re treat for simple warming perturbations are described. One important result is that much larger grounding line retreat is obtained with a Coulomb-friction based parameterization of grounding line flux, compared to that based on power-law sliding, but further testing may be desirable (see below).

**I would like to thank the referee for this early and very detailed review, which gave me ample time to check out in more detail the concerns that were raised. Here, I will answer to the major questions raised by the referee and how new implementations hopefully improved the model and the model results.**

Specific comments:

(1) The treatment of ice temperatures is based on classic vertical profile equilibrium solutions which allow for vertical ice velocity, and then time lagged with an e-folding relaxation towards these solutions at each grid point. The timescale of the e-folding lag is based reasonably on the local Peclet number (pg. 17, eq. 42). This is probably the most drastic simplification from other 3-D hybrid models, and neglects horizontal advection (which cools mid-level interiors as cold surface ice is advected downwards and outwards, and cools the cores of ice shelves supplied by flow across thick grounding lines. A fairly arbitrary compensation for this lack of cooling is attempted by reducing the strain heating (pg. 17, line 6). This simplified temperature treatment is evident in the benchmark intercomparisons in the Appendices, where basal temperature is the only field with poor results.

As a suggestion, perhaps basic horizontal temperature advection could be added to the model, ust by adding an additional term in Eq. (41): ... + u dT/x + v dT/dy with (u,v) given by (12) and T is the column mean temperature. That probably would not require much CPU or slowdown of the model.

**The referee is correct that this is probably a drastic simplification. It was also one of the first simplifications I made to the model. However, my major concern was not so much omitting horizontal advection, which it is relatively well counterbalanced by frictional heating, as the EISMINT I experiments show. I admit that this has not so much of a physical basis and the coupled experiments in EISMINT II were not very convincing. My major concern with this approximation is related to the time-dependent evolution of the temperature, which in its current form is not suited for paleo-climatic studies. Therefore, I revised the temperature calculation completely by solving the time-dependent thermodynamic equation in three dimensions, similar to Pattyn (2010). It includes besides vertical diffusion and advection also horizontal advection, internal heating and frictional heating. In order to improve calculation speed, the whole subroutine was optimized and given the stability of the numerical scheme, the temperature field is only updated every 10 to 20 iterations.**

**Nevertheless, there are a couple of simplifications made: (i) the temperature field is calculated using shape functions for both horizontal and vertical velocity (Hindmarsh, 1999) as well as for velocity gradients based on the deformational SIA velocity for a vertically-integrated value of A; (ii) the flow parameter A is still determined for a given column fraction. Despite these simplifications, the model is now in agreement with both the EISMINT I (Huybrechts et al, 1996) and EISMINT II (Payne et al, 2000) experiments. Even the 'unstable' basal temperature patterns according to some experiments in EISMINT II are now reproduced.**

Given this concern, I suggest that a map of the models basal temperatures for modern Antarctica be shown, and compared with existing model and data based maps (of
which the author is a leader).

**It was my mistake not to have shown the basal temperature field for the Antarctic ice sheet, especially since the temperature calculation was a major approximation. The basal temperature field was different from the one given in Pattyn (2010), as it does not include the optimization of geothermal heat flow using observed temperature profiles and subglacial lake distribution. However, it was more in line with other basal temperature fields obtained by other model studies (Pollard and DeConto; Huybrechts, ...), which gave me confidence in the approximation. A comparison between the new basal temperature field and the approximation of the submitted manuscript reveals also the same pattern, which demonstrates that the initial approximation was quite well representing the ice-sheet temperature field. It is therefore expected that the forcing experiments using the new temperature calculation will not be so different from the previous ones, especially on the time scales that are considered. Nevertheless, the concerns on the temperature field are now taken away.**

(2) It is puzzling why the inverse procedure for basal sliding coefficients (p. 23-24, Fig. 5) yields quite large errors in surface elevation (âĹij200 m) in some regions of the interior East Antarctic plateau. The inverse procedure should reduce them to 10's m (Pollard and DeConto, 2012b) (even if the bed elevations are in error, model or observed, cf. pg. 24 line 19).

**Thanks for remarking this. I have been looking into this in more detail. First of all, it seems that better convergence is reached when letting the optimization run for 200 ka instead of 100 ka. Secondly, the use of the regularization term (smoothing) improves the fit near the borders (compared to Pollard and DeConto, 2012b), but increases the error in the interior. Therefore, the global fit improves, but the referee is right to point that the regularization results in a poorer match for the interior ice sheet. I should state it otherwise.**
Perhaps these larger errors are due to regions of the bed erroneously being frozen. In frozen basal regions the inverse procedure cannot reduce the model's surface elevation errors. So this is an additional reason to request a basal-temperature map.

**Some errors are due to frozen zones, since the optimization procedure does not perform across these zones. However, as shown by the similarity between the basal temperature fields (old and new), this seems not due to a mismatch of frozen/temperate areas.**

nb: "ice thickness", pg.24 line 1, should probably be "ice surface elevation".

**Indeed it is.**

(3) One important result is the greater grounding line retreat with TGL (Coulomb-friction based grounding line flux parameterization, Eq. 25), vs SGL (power-law sliding based, Eq. 23). All experiments shown use power-law sliding (Eq. 15) for the interior grounded ice, and none use Coulomb sliding (Eq. 21). My concern is that the combination of TGL with power-law interior sliding is not compatible, and the mismatch in the physics may lead to spurious behavior in grounding zone regions. (The discussion on pg. 13, lines 24-27 may be relevant).

To address this concern, I would request additional runs be made with Coulomb friction law (Eq. 21) and the TGL grounding line parameterization. This would ideally also involve re-doing the optimization spin-up for basal properties, which may still be feasible by changing phi (till friction angle) instead of $A_b in Eq.(55). Alternatively, the combined Eq.(22) could be used instead of(21).$

**This has been looked into with greater detail. First of all, I don't completely agree with the non-compatibility between the Coulomb boundary condition at the grounding line and the Weertman sliding law inland from it. As shown in Tsai et al (2015), the Coulomb friction leading to vanishing effective pressure at the grounding line is a physically correct condition, whilst with the power law,**

**where the effective pressure is non-zero at the grounding line. Furthermore, the crossover from Coulomb conditions at the grounding line to power-law conditions inland is a very narrow transition zones (with exception of perhaps the Siple Coast region where streams experience a very low drag for a wide area). The contact with the ocean will always be influenced by marine sediments (characterized by a till friction angle), which makes the combination of both conditions (power law sliding for the ice sheet and Coulomb friction for the grounding line) valid. To demonstrate this, I carried out different experiments with varying values of till friction angle at the grounding line. Only for high till friction angles $\phi > 50°$) does the grounding-line sensitivity diminish, but still remains more sensitive than the grounding line conditions according to the power-law sliding. Moreover, I also included an optimization scheme for the Coulomb friction law (on the suggestion by the referee). This optimization changes tan(phi) (and not phi as the referee suggested). The resulting fit is less well than with the power law, but it makes phi vary between 2- 70°. Higher/lower values would be really non-physical. The resulting response is obviously less sensitive than with the one where phi is prescribed, but still more sensitive than the power-law sliding and Schoof-condition at the grounding line. Both results are interesting and will be discussed in detail in the paper.**

(4) The use of driving stress instead of basal stress in the basal sliding law to avoid iterations (pg. 10, Eqs. 15,16) is one of the features used to speed up the model. But maybe the 20

**Basal sliding with the hybrid model IS a function of basal shear stress (or basal drag). So this effect of driving stresses being balanced by driving stresses is not correct (I should write this better in the revised manuscript). The equations are correct for $m = 1$. However, for a power law with $m = 2$, for instance, Eqs 15,16 make the sliding law more viscous than plastic by introducing the term $tau_d$. However, the revised model now properly calculates the effective viscosity in**

the SSA equations (see response to referee 2), hence requiring iteration, so that this approximation is not made anymore. The resulting effect is rather limited, as I expected.

(5) The subglacial water pressure $p_w$ in Eqs. (19) and (20), pg. 11, is assumed to depend on elevation minus sea level, which is a common step in many models. But it is hard to see how the subglacial water system can sense hydrostatic pressure from the ocean at all, more than 100 or 200 km inland from the grounding line.

**I know. That is exactly why I did only use the Coulomb condition at the grounding line, because here the effective pressure is zero by definition. Given the fact that I now have introduced the optimization of the Coulomb friction law for the interior ice sheet, the approximate definition of $p_w$ can be seriously questioned (which I will discuss). As I already mentioned in the manuscript, a subglacial hydrology model would be more appropriate and physically correct.**

Technical points:

**Will be answered with the new version of the manuscript.**

---

## Author Comment (AC2) · 10 Apr 2017

**Interactive comment on "Sea-level response to melting of Antarctic ice shelves on multi-centennial time scales with the fast Elementary Thermomechanical Ice Sheet model (f.ETISh v1.0)" by Frank Pattyn**

S. L. Cornford (Referee)

This manuscript presents the details and some experiments carried out with a new ice

sheet model code (f.ETISH), which is designed to represent the major process in near future ice sheet dynamics (e.g changes in grounding line flux due to ice shelf thinning) well enough to be meaningful but not requiring fine spatial resolution and the attendant computational cost. The new model is similar in that respect to the model of Pollard and DeConto (sans cliff collapse), but makes some additional approximations for the sake of speed. Is this a useful new model? Perhaps. It could be very well suited to long-term integrations, though that is equally true of Pollard and DeConto. It could be useful in large ensemble construction, where I think it makes a better job of representing the physics than (say) Ritz et al 2015. It does seem to contain new but not really well justified approximations in rheology and temperature structure, but given that these things are largely unknown and must be tuned anyway, that may not be very serious problem. The paper is a bit rambling in parts (maybe this review is too), especially the model description and the discussion. Sometimes it includes expressions that are well known, then ignores them, e.g the effective viscosity is treated this way, and the discussion of Coulomb friction laws seem to be a bit extraneous too. I think it really does need a substantial review and edit.

**I should like to thank Stephen for this thorough review and I will try to give a non-rambling response to his queries. A series of improvements to the model have been made and are detailed in my response to Referee 1. I will therefore refer to my response to that referee for points that were already raised. The major changes to the model are (i) the full temperature calculation, including horizontal advection and internal heating, (ii) an optimization scheme for the Coulomb friction law (and therefore tests on different values of $\phi$ at the grounding line, (iii) the SSA solution based on a properly calculated effective viscosity (instead of a crude approximation).**

General Comments

One message of the paper seems to be that treating the flux across the grounding line according to Tsai (TGL) rather than Schoof (SGL) dramatically increases the retreat

rate. This could be because Tsai depends on a higher power of flotation thickness (more acceleration of retreat on retrograde slopes), but might to some extent be attributed to the addition of another free parameter (tan phi) . Given that the model is so quick, maybe some runs with larger tan($\varphi$)? Looking at eq 18, for much of WAIS where bedrock elevation, is < -1 km, tan($\varphi$) is around tan($\varphi$min) = 0.2. What happens if tan($\varphi$) = tan($\varphi$max) ≈ 0.5 – presumably we see about half the rate of SLR? On the same note, TGL is double sided - flux increases more quickly with grounding line thickness, which would mean that the grounding line accelerates more readily in unstable configurations (retrograde slopes, without buttressing) but decelerates more readily in stable configurations (prograde slopes) - and the formula for tan(phi) should amplify this effect somewhat. The East Antarctic results in section 5 seem to differ from this, with the introduction of TGL leading to retreat over prograde slopes (Totten, Wilkes Basin, Recovery Glacier a little upstream from the present day GL) where there was little in the results with SGL.

**See my response to Referee 1. Higher values of $\phi$ lead to lower sensitivity of grounding line retreat, but it is not half the amount of SLR for a doubling of $\phi$. Even for very high values of $\phi > 70°$, this leads still to a mass loss that is significantly higher than for the Schoof-condition. I included also an optimization of the Coulomb friction law, whereby values of till friction angle are optimized, hence also at the grounding line. There will be a more thorough discussion on the retreat experiments for different conditions of $\phi$ and the sensitivity on prograde slopes in the revised manuscript.**

I am suspicious of approximating the effective viscosity by assuming that the stress that enters it is that of an free floating 1HD shelf (eqs 8 and 9). Can this really be a good approximation in buttressed ice shelves like (e.g) PIG, Amery, Totten, where there are regions with little along flow stretching, but strong lateral strains ? To me, the example in appendix D is not especially convincing – cross flow gradients are too low, and the flow field too smooth. This might not be a big error in itself, since at higher melt rates

all that will matter is the TGL / SGL with no buttressing, but a more convincing test is needed. Why not re-run a middle melt-rate experiment with the normal nonlinear rheology?

**I have been looking into this and calculated the effective viscosity now as it should be (according to eq. 5). It also required an iteration, but this doesn't seem to slow down the model as much. The resulting effective viscosity is different (as would be expected), and in few cases the response as well, mostly related to differences in buttressing factors. However, the major sensitivity of the model still remains with the treatment of the boundary conditions at the grounding line (Tsai vs Schoof), and the magnitude of change is comparable to the results presented in the submitted manuscript. A new set of results will therefore be presented in the revised manuscript.**

Given that there is a well known test - MISMIP+ - with published results that include both the Tsai friction rule and ice shelf buttressing - why not test f.ETISH against that?

**It is of course an interesting idea that will require quite some work and also fall outside the scope of the present paper. In term, it is envisaged to perform those tests, but it will require major changes in the model code with respect to the adaptability of the boundary conditions.**

Specific comments

**These will be answered in detail with the revised manuscript.**

---

## Author Response (AR1)

**Rebuttal: 'Sea-level response to melting of Antarctic ice shelves on multi-centennial time scales with the fast Elementary Thermomechanical Ice Sheet model (f.ETISh v1.0)' by Frank Pattyn**

**I would like to thank both referees for their thorough assessment of my submitted paper. Thanks to their thoughtful comments, I made several improvements to the model and reran the experiments again. I also added extra experiments in line with the concerns of the referees. The experiments on millennium time scales were omitted, because they did not add anything significant to the paper that would become needlessly too long otherwise. In view of the model changes, the appendices have been adapted as well. The major model changes are:**

- **The complete temperature field is now calculated (based on SIA shape functions for the different advection terms). Therefore, the temperature field includes now horizontal advection and is also correctly time-dependent. Thermomechanical coupling has not been altered (2d coupling)**

- **The effective viscosity in the SSA equations has not been approximated, but is now iteratively solved. As a consequence, basal drag is not linearized (made function of driving stress) any longer.**

- **The Coulomb friction law has been used in conjunction with the TGL flux condition at the grounding line, to make it coherent. This means that the optimization procedure has been adapted and friction coefficients (tan(phi)) are optimized during initialization.**

**1 Anonymous Referee 1**

**1.1 General comments:**

This paper presents a thorough and clear description of a new ice sheet model, akin to hybrid dynamics SIA/SSA models currently used for Antarctica, but with some reasonable and innovative simplifications so it is computationally

fast. The model is implemented in MATLAB and will be a useful tool to engage students in teaching and workshop environments, as well as being capable for many research applications.

In this paper the model is thoroughly tested against established benchmarks (EISMINT, MISMIP) and validated vs. modern Antarctica. Sensitivity experiments of Antarctic re treat for simple warming perturbations are described. One important result is that much larger grounding line retreat is obtained with a Coulomb-friction based parameterization of grounding line flux, compared to that based on power-law sliding, but further testing may be desirable (see below).

**I would like to thank the referee for this early and very detailed review, which gave me ample time to check out in more detail the concerns that were raised.**

**1.2 Specific comments:**

(1) The treatment of ice temperatures is based on classic vertical profile equilibrium solutions which allow for vertical ice velocity, and then time lagged with an e-folding relaxation towards these solutions at each grid point. The timescale of the e-folding lag is based reasonably on the local Peclet number (pg. 17, eq. 42). This is probably the most drastic simplification from other 3-D hybrid models, and neglects horizontal advection (which cools mid-level interiors as cold surface ice is advected downwards and outwards, and cools the cores of ice shelves supplied by flow across thick grounding lines. A fairly arbitrary compensation for this lack of cooling is attempted by reducing the strain heating (pg. 17, line 6). This simplified temperature treatment is evident in the benchmark intercomparisons in the Appendices, where basal temperature is the only field with poor results.

As a suggestion, perhaps basic horizontal temperature advection could be added to the model, ust by adding an additional term in Eq. (41): ... + u dT/x + v dT/dy with (u,v) given by (12) and T is the column mean temperature. That probably would not require much CPU or slowdown of the model.

**The referee is correct that this is probably a drastic simplification. It was also one of the first simplifications I made to the model. However, my major concern was not so much omitting horizontal advection, which it is relatively well counterbalanced by frictional heating, as the EISMINT I experiments show. I admit that this has not so much of a physical basis and the coupled experiments in EISMINT II were not very convincing. My major concern with this approximation is related to the time-dependent evolution of the temperature, which in its current form is not suited for paleo-climatic studies. Therefore, I revised the temperature calculation completely by solving the time-dependent thermodynamic equation in three dimensions, similar to Pattyn (2010). It includes besides vertical diffusion and advection also horizontal advection, internal heating and frictional heating. In order to improve calculation speed, the whole subroutine was optimized and given the stability of the numerical scheme, the temperature field is only updated every 10 to 20 iterations.**

**Nevertheless, a couple of simplifications remain: (i) the temperature field is calculated using shape functions for both horizontal and vertical velocity (Hindmarsh, 1999) as well as for velocity gradients based on the deforma-**

**tional SIA velocity for a vertically-integrated value of A; (ii) the flow parameter A is still determined for a given column fraction. Despite these simplifications, the model is now in agreement with both the EISMINT I (Huybrechts et al, 1996) and EISMINT II (Payne et al, 2000) experiments. Even the 'unstable' basal temperature patterns according to some experiments in EISMINT II are now reproduced (and more results are presented in the Appendix).**

Given this concern, I suggest that a map of the models basal temperatures for modern Antarctica be shown, and compared with existing model and data based maps (of which the author is a leader).

**It was my mistake not to have shown the basal temperature field for the Antarctic ice sheet, especially since the temperature calculation was a major approximation. The basal temperature field was different from the one given in Pattyn (2010), as it does not include the optimization of geothermal heat flow using observed temperature profiles and subglacial lake distribution. However, it was more in line with other basal temperature fields obtained by other model studies (Pollard and DeConto; Huybrechts, ...), which gave me confidence in the approximation. A comparison between the new basal temperature field and the approximation of the submitted manuscript reveals also the same pattern, which demonstrates that the initial approximation was quite well representing the ice-sheet temperature field. The basal temperature field is now included as a figure in the manuscript.**

(2) It is puzzling why the inverse procedure for basal sliding coefficients (p. 23-24, Fig. 5) yields quite large errors in surface elevation (200 m) in some regions of the interior East Antarctic plateau. The inverse procedure should reduce them to 10's m (Pollard and DeConto, 2012b) (even if the bed elevations are in error, model or observed, cf. pg. 24 line 19).

**Thanks for remarking this. I have been looking into this in more detail. It seems that the use of the regularization term (smoothing) improves the fit near the borders (compared to Pollard and DeConto, 2012b), but increases the error in the interior. I adapted the regularization scheme so that it only applies for marine boundaries (bedrock below sea level). This way the fit is better in the interior as well as close to the boundaries. This has also been stated in the manuscript.**

Perhaps these larger errors are due to regions of the bed erroneously being frozen. In frozen basal regions the inverse procedure cannot reduce the model's surface elevation errors. So this is an additional reason to request a basal-temperature map.

**Some errors are due to frozen zones, since the optimization procedure does not perform across these zones. However, as shown by the similarity between the basal temperature fields (old and new), this seems not due to a mismatch of frozen/temperate areas.**

nb: "ice thickness", pg.24 line 1, should probably be "ice surface elevation".

**Indeed it is. Corrected.**

(3) One important result is the greater grounding line retreat with TGL (Coulomb-friction based grounding line flux parameterization, Eq. 25), vs SGL (power-law sliding based, Eq. 23). All experiments shown use power-law sliding (Eq. 15) for the interior grounded ice, and none use Coulomb sliding (Eq. 21). My concern is that the combination of TGL with power-law interior sliding is not compatible, and the mismatch in the physics may lead to spurious behavior

in grounding zone regions. (The discussion on pg. 13, lines 24-27 may be relevant).

To address this concern, I would request additional runs be made with Coulomb friction law (Eq. 21) and the TGL grounding line parameterization. This would ideally also involve re-doing the optimization spin-up for basal properties, which may still be feasible by changing phi (till friction angle) instead of $A_b$ in Eq. (55). Alternatively, the combined Eq. (22) could be used instead of (21).

**This has been looked into with greater detail. First of all, I don't completely agree with the non-compatibility between the Coulomb boundary condition at the grounding line and the Weertman sliding law inland from it. As shown in Tsai et al (2015), the Coulomb friction leading to vanishing effective pressure at the grounding line is a physically correct condition that can be used independently of the basal characteristics of the inland ice sheet. The crossover from Coulomb conditions at the grounding line to power-law conditions inland is a very narrow transition zones (with exception of perhaps the Siple Coast region where streams experience a very low drag for a wide area). The contact with the ocean will always be influenced by marine sediments (characterized by a till friction angle), which makes the combination of both conditions (power law sliding for the ice sheet and Coulomb friction for the grounding line) valid. To demonstrate this, I carried out different experiments with varying values of till friction angle at the grounding line. Only for high till friction angles $\phi > 50°$) does the grounding-line sensitivity diminish, but still remains more sensitive than the grounding line conditions according to the power-law sliding. Moreover, I also included an optimization scheme for the Coulomb friction law (on the suggestion by the referee). This optimization changes tan(phi) (and not phi as the referee suggested). The resulting fit is less well than with the power law, but it makes phi vary between 2- 70°. Higher/lower values would be really non-physical. The resulting response is obviously less sensitive than with the one where phi is prescribed, but still more sensitive than the power-law sliding and Schoof-condition at the grounding line. As a comparison, the combination of Weertman sliding and TGL condition are also presented. This reveals that the most dominant factor in the sensitivity is the TGL condition, not the type of sliding/friction inland.**

(4) The use of driving stress instead of basal stress in the basal sliding law to avoid iterations (pg. 10, Eqs. 15,16) is one of the features used to speed up the model. But maybe the 20% of the ice sheet where driving stresses are not essentially balanced by basal stresses (p.10, lines 16-17) are in important regions such as ice streams. This concern could be addressed by one sensitivity test in which the approximation in Eq. (16) is not made (requiring expensive iteration).

**Basal sliding with the hybrid model IS a function of basal shear stress (or basal drag). So this effect of driving stresses being balanced by driving stresses is not correct. The equations are correct for $m = 1$. However, for a power law with $m = 2$, for instance, Eqs 15,16 make the sliding law more viscous than plastic by introducing the term $tau_d$. However, the revised model now properly calculates the effective viscosity in the SSA equations (see response to referee 2), hence requiring iteration, so that this approximation is not made anymore.**

(5) The subglacial water pressure $p_w$ in Eqs. (19) and (20), pg. 11, is assumed to

depend on elevation minus sea level, which is a common step in many models. But it is hard to see how the subglacial water system can sense hydrostatic pressure from the ocean at all, more than 100 or 200 km inland from the grounding line.

**I know. That is exactly why I did only use the Coulomb condition at the grounding line, because here the effective pressure is zero by definition. Given the fact that I now have introduced the optimization of the Coulomb friction law for the interior ice sheet, the approximate definition of $p_w$ can be questioned, but has also been used by several other authors. As I already mentioned in the manuscript, a subglacial hydrology model would be more appropriate and physically correct.**

Technical points:

p.3, Fig. 1. I suggest indicating in the figure that sea level is at $z = 0$, as seems to be required in Eqs. 18, 19 and 20. And $z_{sl}$ must $= 0$ (p.7, line 3). Alternatively, replace $b$ throughout p.11 with $b - z_{sl}$.

**I replaced $b$ by $b - z_{sl}$ as suggested by the referee**

p.7, Eq (2). More correctly, $v_{sia} = v_b + ...|\tau_d|^{n-1}\tau_d$ p.9, line 7 et seq. To avoid confusion, say explicitly that $\tau_f$ is the free-floating stress, used later in Eq. (24) as well as in (3),(4) via eta in (11).

**Corrected**

p.11, line 3: Why might the friction angle phi be a function of bedrock elevation, physically? p.12, lines 6-7. The sentence " However, expressed as a ..." is unclear to me.

**This sentence has been removed.**

p.13, lines 7-10. Here, it might be helpful to mention that a staggered grid (Arakawa C) grid is used as shown in Fig. 2.

**Done**

p.14, Eq. (27). Say that this is only applicable for SIA advection.

**Done; I explicitly wrote that for the grounded ice sheet according to the SIA model this equation is written as a diffusion equation for ice thickness.**

p.14, lines 11-22: Say whether this 'maximum strain check' is applied everywhere, on ice shelves, or just at the grounding line.

**I wrote that this is checked at the grounding line.**

p.17, line 24 and Eq.(42). Say that this is vertical advection (not horizontal).

**This equation has been removed due to the changes in the temperature calculation.**

p.18, line 13. Specify the value of $E_f$ used for ice shelves.

**Done. It is 0.5.**

p. 19, lines 1 and 10. Say that the equilibrium bed topography and loads ($b_{eq}$, $h_{eq}$, $hw_{eq}$) are taken from modern observed fields (Bedmap2), if that is the case.

**Done**

p.19, line 18.Say that the local numerical integration is for Eq.(38) (I think).

**This equation has been removed due to the changes in the temperature calculation.**

p.19, line 24. Iterations are also eliminated due to the approximation of driving = basal stress in Eq. (16).

**This doesn't apply anymore.**

p.21, line 24. A simple one-valued PDD is ok for modern Antarctica with little surface melt. But the surface melt treatment will need improving (snow vs.

ice, refreezing, etc.) to represent greatly increased surface melt around the Antarctic margins in warm future climates.

**I agree. A similar approach was adopted by Pollard and DeConto (2012). However, on the time scales I consider, surface melt has not a decisive impact. Refreezing would lead to a higher retention of the melt water, hence limit the impact of surface melt even more. It will certainly become more important if other ice shelf disintegration processes, such as hydro-fracturing, would be taken into account.**

p.22, line 9. Say where ocean temperatures $T_{oc}$ are obtained from. Actually it seems that Eq. (53) and $T_{oc}$ are not used in any experiments here, for which the melt rate M is simply prescribed region by region (p.30, lines 17-19).

**This section has been removed. It is mentioned in the manuscript that the mechanism is included in f.ETISh, but that only constant values of melt are applied to the ice shelves in this paper.**

p.22, line 16. Eq. (53) produces higher melt rates closest to the grounding line not because it's quadratic, but because the freezing temperature $T_f o$ decreases with depth (noting $h_b$ in Eq. (54) is negative below sea level).

**This section has been removed (see above).**

p.23, line 1. Perhaps change "further constrained by" to "driven by".

**Done.**

p.25, line 14. Change "back by" to "back to". p.27, line 3. Perhaps change to "of the model and the approximations..."

**Done.**

p.28, Fig. 9 caption, 2nd line. Remove "(a)".

**Done. Replaced by 'years' to make clear what the units are.**

p.30, line 8. Perhaps change "provoked" to "applied".

**Done.**

p.30, line 10. *Why* are TGL runs characterized by higher driving stresses?

**Referee 2 made a whole point of the driving stresses and I agree that the reasoning should be the other way around. The origin of the high driving stresses is that for a same ice flux, surface gradients must be larger with TGL. See rebuttal to referee 2 for more details. The figure of driving stresses has been removed.**

p.31, Fig. 11 caption, 2nd line. Say "Atmospheric temperature forcing is..."

**Done.**

p.34, line 22. Mention that the agreement with the benchmark(s) is shown in the Appendices.

**Done.**

p.34, lines 27-28. Change to "Despite the approximations, the results of thermomechanical coupling of ice sheet flow are also in good agreement with the EISMINT benchmark..."

**This has been changed, since the thermodynamical part of the model has been improved.**

pg.34, line 29: Perhaps change to "compared to the other benchmarks,"

**Rephrased.**

pg.35, lines 7-8. Change to "requires membrane stresses at both sides of the grounding line to be resolved with sufficient detail (Schoof 2007a),"

**Done.**

pg.35, line 10. Mention in parentheses that this rule is that on pg. 13, lines 9-10.

**I made a reference to the specific section.**
pg.35, lines 28-29. Change to "Direct comparison is not possible with recent studies of Antarctic ice mass loss that are forced by atmosphere-ocean models following socalled RCPs (Representative Concentration Pathways). Direct comparison with the SeaRISE...
**Done.**
pg.36, line 8. Perhaps describe in a few words what "RCP8.5 amplification" is.
**This paragraph in the discussion has been deleted, since the millennium time scales have been left out.**
pg.36, lines 28-29. Perhaps change to "with the exception of grounding-line zones with small-scale bedrock variability, where grounding-line response to atmospheric and oceanic forcing is sensitive to spatial resolution."
**Done.**
pg.37, line 8. The "dominance of ocean forcing" in this paper relies on the absence of physics such as hydrofracturing that occur due to large increases in surface melting around the margins. With RCP8.5 at least, there will be a huge increase in the latter within ?100 to 200 years, which could affect the ice sheet in unexpected ways.
**Indeed. This has been rephrased and at several places in the manuscript it has been mentioned that hydro-fracturing (as atmopsheric forcing) may induce a much larger impact.**
Appendices, figure captions A2 to A5. It would help to specify the benchmark experiment (EISMINT I or II, MISMIP, etc) in each caption, especially if the figures appear on different pages than the relevant text.
**Done.**

**2   S. L. Cornford (Referee)**

This manuscript presents the details and some experiments carried out with a new ice sheet model code (f.ETISH), which is designed to represent the major process in near future ice sheet dynamics (e.g changes in grounding line flux due to ice shelf thinning) well enough to be meaningful but not requiring fine spatial resolution and the attendant computational cost. The new model is similar in that respect to the model of Pollard and DeConto (sans cliff collapse), but makes some additional approximations for the sake of speed. Is this a useful new model? Perhaps. It could be very well suited to long-term integrations, though that is equally true of Pollard and DeConto. It could be useful in large ensemble construction, where I think it makes a better job of representing the physics than (say) Ritz et al 2015. It does seem to contain new but not really well justified approximations in rheology and temperature structure, but given that these things are largely unknown and must be tuned anyway, that may not be very serious problem. The paper is a bit rambling in parts (maybe this review is too), especially the model description and the discussion. Sometimes it includes expressions that are well known, then ignores them, e.g the effective viscosity is treated this way, and the discussion of Coulomb friction laws seem to be a bit extraneous too. I think it really does need a substantial review and edit.

I should like to thank Stephen for this thorough review and I will try to give a non-rambling response to his queries. A series of improvements to the model have been made and are detailed in my response to Referee 1. I will therefore refer to my response to that referee for points that were already raised and briefly mentioned in my introduction to the rebuttal. Given the changes, the model description has been shortened as well, and unnecessary items have been discarded.

**2.1 General Comments**

One message of the paper seems to be that treating the flux across the grounding line according to Tsai (TGL) rather than Schoof (SGL) dramatically increases the retreat rate. This could be because Tsai depends on a higher power of flotation thickness (more acceleration of retreat on retrograde slopes), but might to some extent be attributed to the addition of another free parameter (tan phi) . Given that the model is so quick, maybe some runs with larger tan(phi)? Looking at eq 18, for much of WAIS where bedrock elevation, is < -1 km, tan(phi) is around tan(phimin) = 0.2. What happens if tan(phi) = tan(phimax) = 0.5 - presumably we see about half the rate of SLR? On the same note, TGL is double sided - flux increases more quickly with grounding line thickness, which would mean that the grounding line accelerates more readily in unstable configurations (retrograde slopes, without buttressing) but decelerates more readily in stable configurations (prograde slopes) - and the formula for tan(phi) should amplify this effect somewhat. The East Antarctic results in section 5 seem to differ from this, with the introduction of TGL leading to retreat over prograde slopes (Totten, Wilkes Basin, Recovery Glacier a little upstream from the present day GL) where there was little in the results with SGL.

**See my response to Referee 1. Higher values of $\phi$ lead to lower sensitivity of grounding line retreat, but it is not half the amount of SLR for a doubling of $\phi$. Even for very high values of $\phi > 70°$, this leads still to a mass loss that is significantly higher than for the Schoof-condition. I added this in the text. I included also an optimization of the Coulomb friction law, whereby values of till friction angle are optimized, hence also at the grounding line. All experiments with the TGL condition are now run with the Optimized Coulomb friction law. For the 'ice-shelf removal' experiment, the combination Weertman sliding-TGL was added as a comparison, showing that the higher sensitivity is clearly related to TGL. In that section, it is explained in detail what the origin of the difference is (related to the remarks of the referee on driving stresses).**

I am suspicious of approximating the effective viscosity by assuming that the stress that enters it is that of an free floating 1HD shelf (eqs 8 and 9). Can this really be a good approximation in buttressed ice shelves like (e.g) PIG, Amery, Totten, where there are regions with little along flow stretching, but strong lateral strains ? To me, the example in appendix D is not especially convincing - cross flow gradients are too low, and the flow field too smooth. This might not be a big error in itself, since at higher melt rates all that will matter is the TGL / SGL with no buttressing, but a more convincing test is needed. Why not re-run a middle melt-rate experiment with the normal nonlinear rheology?

**I calculated the effective viscosity now as it should be (according to eq. 5).**

**It also required iteration, but this doesn't seem to slow down the model as much. The resulting effective viscosity is different (as would be expected), and in few cases the response as well. However, the major sensitivity of the model still remains with the treatment of the boundary conditions at the grounding line (Tsai vs Schoof), and the magnitude of change is comparable to the results presented in the initial manuscript.**

Given that there is a well known test - MISMIP+ - with published results that include both the Tsai friction rule and ice shelf buttressing - why not test f.ETISH against that?

**It is of course an interesting idea that will require quite some work and also falls outside the scope of the present paper. In term, it is envisaged to perform those tests, but it will require major changes in the model code with respect to the adaptation of the boundary conditions.**

**2.2 Specific comments**

**2.2.1 Abstract**

(and elsewhere) 'The higher sensitivity [in the case of the Tsai 2] is attributed to higher driving stresses upstream from the grounding line.' I'm not sure this makes sense - and I suggest it is at least partly the other way round. Because q(TGL) is larger than q(SGL), but both are only applied at the GL, dh/dx is going to be bigger at the GL for TGL with all else being equal. The same - plain Weertman - friction law is applied upstream.

**I corrected this (here and elsewhere in the manuscript). The higher sensitivity is attributed to higher ice fluxes at the grounding line due to vanishing effective pressure.**

**2.2.2 Section 1**

'The majority of these interactions demonstrate non-linear behaviour due to feedbacks, leading to self-amplifying ice mass change.' $- >$ 'Some of these . . .'
**Corrected.**

'thicker ice grounded in deeper water would result in floatation, increased ice discharge, and further retreat within a positive feedback loop.' $- >$ thicker ice grounded in deeper water would result in increased ice discharge, and further retreat within a positive feedback loop.'
**Corrected.**

. .. based on boundary layer theory (...Ritz et al., 2015. . .). I don't think the Ritz et al., 2015 GL is based on boundary layer theory, does it? But imposes retreat rates sampled from some sort of probability distribution.
**Indeed, you are right. The use of the BLT in Ritz is not in the same way as in Pollard. I removed the reference.**

**2.2.3 Section 2.1**

'The main advantage of SIA is that the velocity is completely determined from the local ice-sheet geometry.' That might be called the main disadvantage too.
**Well, it makes the computation rather simple for a rather large domain (interior ice sheet) for which this approximation is valid. Anyway, I removed the sentence.**

SSA+SIA : 'a simple addition still guarantees a smooth transition' - why wouldn't it? SIA isn't smooth in the same way as SSA, but so long as the surface eleva­tion is smooth, it will be. More to the point, is this a good approximation? How about at the ice shelf calving front, where grad(s) is large, there is no basal stick and SIA makes no sense? I don't think Schoof and Hindmarsh 2010 gives us a reason to think that SSA+SIA is any more sensible than plain SSA.

**I don't understand this quite well. The addition of both is done for the grounded ice sheet alone. The difference with the Schoof and Hindmarsh approach is that the effective strain doesn't take into account vertical shear­ing In the ice shelves, the SIA velocity is always kept zero (no shearing). I made this clear to avoid confusion.**

'Basal velocities in the hybrid model are defined through a friction power law, where' Basal traction, no? The velocity is related but depends also on viscous stress at least close to the GL in the SSA+SIA case.

.

2..1.7. The Coulomb friction law plays no part in the results, except for its involvement with the Tsai flux. I suggest cutting this (longish) section 2.1.5 entirely and describing tan(phi) in 2.1.7

**This has been changed. The experiments are now run with the Coulomb friction law as well. I therefore kept this section in. Given the changes in the model, the description is shortened elsewhere anyway.**

'where 'spy' is the number of seconds per year' Why switch units mid-expression? Eq 25: What value does tan(phi) tend to take?

**I removed this. The units are everywhere added when appropriate.**

**2.2.4   Section 2.2**

eq 28: should be an inequality? hdu/dx ≤ AhTf (ie the maximum stretching is in free shelves) 'Ritz et al. (2015) use a slightly different prescription, but sensi­tivity tests showed that the extra terms in the mass conservation equation can be safely dropped, rendering the maximum strain check therefore indepen­dent of velocity gradients.' Which terms? The terms that have been dropped are ?udh/dx and ?vdh/dx, both of which involve thickness gradients and are typically positive, so in fact dh/dt ≥ a ? ?M ? ?h(du/dx + dv/dy) ≤ a ? M ? 2AhTf And you assume dh/dx is neglibigle?

**Given the changes made to the SSA description, this condition has been taken in its original form. Therefore, it is only mentioned in the text and not specified in an equation. The reader is referred to Ritz et al (2015).**

'However, to compensate for the absence of horizontal advection in the model, only a fraction fs = 0.25 of the total strain heating amount was added. This value is determined from the EISMINT benchmark experiments (Appendix A).' Should this value not depend at all on ice speed? Is ESIMINT a sufficient test of this quite different dynamics?

**Given the changes in the thermodynamics, this parameter has become su­perfluous.**

**2.2.5   Section 2.3**

OK, these expression come from others. But are they justified in any way.

**2.2.6 Section 2.4**

Eq 33. Has some horizontal advection - needed to eliminate T dw/dz and reduce vertical advection to wdT /dz, but neglects horizontal temperature variation? An even simpler solution might be possible if the conservative advection d/dz(wT) was used and all horizontal transport neglected. Or did I miss something?
**Horizontal advection is now taken into account.**
How is eq 41 based on the Peclet number? So ?t is advection dominated when Pe is large and diffusion dominated when P e ? 0, but does the code actually compute some function of Pe?
**This has become obsolete now.**

**2.2.7 Section 2.6**

Why solve eq 46 with BiCGStab? What preconditioner is used? ILU(0)? UMFpack is MATLAB's default sparse solver, I think, and I guess the matrices are all small (coarse grid), so if there are large ice shelves (so A becomes poorly conditioned) this direct solver might be the better choice (or not)
**The standard sparse Matlab solver works very well, but the use of bicgstab enables to speed up the process. I think that many things can be improved in future on behalf of the numerical solvers. I already optimized the initialization with reduced sparse matrix systems (taking into account only the grid points where ice thickness changes over time). I have a PhD student looking into this matter for the moment.**

**2.2.8 Section 4.**

'This further improves the final fit compared to the non-regularized case..' which is not normally the case with regularization - typically regularization results in worse (or no-better) fit to the observations for the sake of a smoother (or more plausible in some other sense) solution.
**Referee 1 made the same remark. Indeed, it should give worse results. However, it makes the results worse for the interior ice sheet, but improves it for the marine borders. Therefore, I adapted the algorithm so that regularization is applied in the marine sectors only. I modified this in the text.**

**2.2.9 Section 5.1:**

Sorry to bring this up, but Cornford et al 2016, Annals of Glaciology https://doi.org/10.1017/aog.2016.13 does a rather similar experiment (all-Antarctic response to sustained ice shelf removal), with a sub-km model, and the Weertman sliding results could be compared.
**Thank you very much for pointing me to that paper (with the unfortunate typo). Indeed, the results of that paper are somehow comparable to removing ice shelves, although that not all ice shelves are removed (once ice thickness smaller than 100m, the shelves remain). I ran these experiments as well and added the results in the paper under the section 'sub-shelf melting'. I made sure that I corrected the typo and ran the model with the melt rates that were actually used (not the ones that are described in the experimental description). A comparison with the results in Cornford et al (2016)**

**is made at the end of the section of sub-shelf melting. The SGL experiments are quite comparable to this study, albeit that the timing of retreat in the different drainage basins is different in some places. It is explained why.**
Is the rate of SLR labelled incorrectly in fig 10?
**Figure has been removed.**

**2.2.10 Section 5.2**

'Melting is not allowed to be spread out across the grounded part of the 20 ice sheet near the grounding line as is done in some models (Feldmann et al., 2014; Golledge et al., 2015)'. Note that Feldmann and Golledge are not really trying to spread the melting about, they are just applying some melt to finite area grid cells whose centers are grounded but whose neighbours are floating, estimating a floating fraction by interpolating the thickness above flotation. This sounds pretty innocuous - even sensible - in which context the sentence above sounds like the wrong choice. Of course we know it is not the wrong choice, but maybe say something about why?
**I rephrased this: "Melting is only applied to fully floating grid cells, without taking into account the fractional area of grounded grid points that are actually afloat, as done in a few studies (refs)." Whether this is wrong or not, I leave inbetween. I just stated that what the difference is in melt treatment compared to other models.**
'[SLR] determined from the change in ice volume above floatation, hence do not represent the total grounded ice mass loss' Seems like an odd comment - how else would it be computed ? It makes me wonder if the section 5.1 SLR is from total mass loss (indeed the text of section 5.1 suggests that, 'the total mass loss for TGL is three times as large compared to SGL, i.e., a contribution to sea-level rise of 12 m ...'), when I assumed it had been computed from VAF
**I agree it is ambiguous. I corrected this at the different places. I also abbreviated sea-level rise to SLR at the different places in the manuscript.**
Fig 11. Although the 'thick lines (SGL), thin lines (TGL)' plot works for the large delta M, I can't make so well out what is going on at small delta M. how about thin lines with a few symbols (say, circles, squares). Or drop the dM = 10 m/a results?
**It has been made easier now, since fewer experiments have been made.**
Fig 12. To my mind, at least one more grid spacing (there are some runs 16km. right? ) to be able to say much about mesh dependence. You can't test convergence at all with just two, you need to show that results are getting closer to one another as dx → 0
**This is a very good remark. The new series of experiments show that spatial resolution influences the timing of grounding line retreat in the different drainage basins, less so the total amount of SLR (at least over longer periods and for high forcings). However, it is not expected that convergence should be obtained when resolution increases, unless one uses the same bedrock data set at each time (i.e., at high resolution an interpolated bedrock map of the low resolution experiment). Since at higher resolutions the bedrock shows a different variability (small bed rises, for instance), grounding-line migration will be influenced by this. However, having said this, I carried out 4 new experiments with melt perturbations (without dT) at 16km resolution. The results are extremely similar to 25 km, showing that the model**

**has reached convergence. At very high resolutions (<5 km), however, the results could be different for the reason I explained above, but the purpose of the f.ETISh model is to provide model runs at lower spatial resolution that capture the essence in marine ice dynamics.**

**2.2.11 Section 6**

'Another major difference pertains to the marine boundary, with a novel implementation of the grounding-line flux condition according to Tsai et al. (2015), based on a Coulomb friction law (TGL)' 'novel' seems a bit strong, given that Tsai derived the flux formula, and the implementation replaces a very similar formula (SGL) in an overall method to modify the Schoof flux to include buttressing due to Pollard.

**I rephrased this.**

p35 'unless sub-grid grounding-line parametrizations are used that generally allow for grid sizes of 10 km (Feldmann et al., 2014).' Personally I think this claim in Feldmann 2014 is not supported by the results, which are better with the sub-grid scheme, but still need dx 1 km. Why should we believe that results in one idealized problem should be widely true?

**True. I removed the claim of 10 km.**

'Nevertheless,comparison with high-resolution SSA and hybrid models show that while differences in transient response exist, results are in overall agreement with the other models (Pattyn and Durand, 2013).' That really was not the message I took from Pattyn and Durand 2013, at least regarding the transient.

**I removed this sentence.**

'as the ice-sheet profiles 'taper off' towards a flattening upper surface, contrary to the power-law case,' - this happens to some extent in the power law case too, depending on the scale length for viscous stresses transmission.

**But for TGL, it is always the case, even for small transition zones.**

'(so-called 'aggressive' grounding line in PISM).' Does Golledge really call it 'agressive' in that paper. I remember him saying it in a talk. Anyway, why not say what it is: a type of numerical error (aggression $-> 0$ as dx $-> 0$) rather than something that could be seen as physics.

**No, Golledge does not call it agressive in the paper, but mentions it in presentations. I removed this now. In any case, this section has been removed since the experiments on longer time spans have been omitted.**

[revised manuscript text omitted]
_{\text{summit}}^3$, where $d_{\text{summit}}$ is defined as $\max(|x - x_{\text{summit}}|, |y - y_{\text{summit}}|)$, expressed in km. Further boundary conditions for the model are zero ice thickness at the edges of the domain and a constant geothermal heat flux of $G = 0.042$ W m$^{-2}$. The ice temperature is not coupled to the ice flow field and a constant value for the flow parameter of $10^{16}$ Pa$^{-n}$ a$^{-1}$ is considered.

The f.ETISh model is a 3d Type I model according to the classification scheme in EISMINT-I, i.e., diffusion coefficients for the grounded ice sheet are calculated on a staggered Arakawa-B grid. Table A1 lists the comparison with data from other 3d Type I models. Both ice thickness and flux compare very well within error bounds of the sample range (limited to only 2–3 models in the  EISMINT-I benchmark, unfortunately). Also the basal temperature at the divide and along the profile is within the limits given by the EISMINT-I benchmark. Differences can be attributed to the use of the shape functions for the velocity field as well as to the use of a staggered grid for the temperature field, whereby the temperature at the divide and along the profile are interpolated values along the central line.

[revised manuscript text omitted]

---

## Author Response (AR2)

**Rebuttal 2**

Dear Hilmar,
Thanks for the very positive news. Below you find the answers to the questions. The corrections are made in the revised manuscript.
Kind regards,
-Frank

Stephen:

I am happy with the revised manuscript, it clearly deals with all the major issues from the first round of reviews.

One bit of pedantry: it is now clear that you have SSA+SIA in the grounded ice and SSA in the shelf, and so the transition is not strictly smooth (in the sense that du/dx, d^2u/dx^2 etc are defined). It is not a big issue in typical ice stream cases (as Winkelmann 2011 notes), but might be in others.

**I have added that this addition is valid for ice stream flow.**

Hilmar:

Top of page 11. : Eq (16) is just the normal viscous sliding law, is it not? Why call it a plastic sliding law? Eq. (16) is clearly not a Coulomb friction law! Are you maybe referring to Eq. (12)?

**Yes, you are right. Thanks for spotting this. Eq. (16) is a sliding law and not a Coulomb friction law. I corrected this. Eq. (12) is the Coulomb friction law.**

Page 12: What controls b_f?

**The user controls b_f. All the experiments in the paper are with either b_f=1 (which is the case presented in Schoof, 2007) or b_f=0, for the experiments where the ice shelves are removed. In the latter case \Theta=1 throughout. Since may be a bit confusing, I corrected the sentence and added another one: $b_f$ is an additional buttressing factor to control the buttressing strength of ice shelves and may be varied between 0 (no buttressing) and 1 (full buttressing). All experiments in this paper use $b_f=1$, except the sensitivity experiments on ice-shelf de-buttressing where $b_f=0$.**

Your \Theta appears different from the one used by Christian. I think you get Christian's definition for b_f=1

**Indeed. See previous remark.**

Do you calculate \tau_{xx} from the model then use (18) to fix u_b? But how do you fix the value of b_f? I saw a statement on page 21 suggesting that b_f is determined from measured surface velocities. Is that correct?

**See previous remark: b_f is a parameter controlled by the user (no real physical meaning). It is not as such determined in the model.**

In (27) I'm missing the shear heating due to deformation in the horizontal plane. What is the justification for not including that part?

**Shear heating due to deformation is given by $\Phi$ in Eq. (27). The thermodynamic model is only based on SIA, so only vertical shear strain heating is included (dv/dz).**

Bit surprise by (29). Thought the Peclet number for ice shelves was >> 1

**The thermodynamic part of the ice shelf is not really crucial for the model, it is merely considered as being a boundary condition. It is a simple analytical solution. Many models just apply a linear profile in the ice shelves (bounded by two Dirichlet conditions given by surface and ocean temperature).**